# SUMOylation of protein phosphatase 5 regulates phosphatase activity and substrate release

Rebecca A Sager[1,2,5], Sarah J Backe[1,2,5], Diana M Dunn [1,2], Jennifer A Heritz [1,2,3], Elham Ahanin [1,2,3], Natela Dushukyan[1,2], Barry Panaretou [4], Gennady Bratslavsky[1,2,3], Mark R Woodford [1,2,3], Dimitra Bourboulia [1,2,3] & Mehdi Mollapour [1,2,3✉]

## Abstract

**The serine/threonine protein phosphatase 5 (PP5) regulates hormone and stress-induced signaling networks. Unlike other phosphoprotein phosphatases, PP5 contains both regulatory and catalytic domains and is further regulated through post-translational modifications (PTMs). Here we identify that SUMOylation of K430 in the catalytic domain of PP5 regulates phosphatase activity. Additionally, phosphorylation of PP5-T362 is pre-requisite for SUMOylation, suggesting the ordered addition of PTMs regulates PP5 function in cells. Using the glucocorticoid receptor, a well known substrate for PP5, we demonstrate that SUMOylation results in substrate release from PP5. We harness this information to create a non-SUMOylatable K430R mutant as a 'substrate trap' and globally identified novel PP5 substrate candidates. Lastly, we generated a consensus dephosphorylation motif using known substrates, and verified its presence in the new candidate substrates. This study unravels the impact of cross talk of SUMOylation and phosphorylation on PP5 phosphatase activity and substrate release in cells.**

**Keywords** Serine-Threonine Protein Phosphatase-5 (PP5); Molecular Chaperone; Co-Chaperone; Heat Shock Protein-90 (Hsp90); SUMOylation
**Subject Categories** Post-translational Modifications & Proteolysis; Signal Transduction; Translation & Protein Quality

## Introduction

Protein phosphatase 5 (PP5) is a serine/threonine protein phosphatase involved in the regulation of cellular functions, including stress response, proliferation, apoptosis, and DNA repair (Golden et al, 2008; Sager et al, 2020). Many regulatory functions of PP5 are a result of its role as a co-chaperone of the molecular chaperone heat shock protein-90 (Hsp90). Unlike other phosphoprotein phosphatase (PPP) family members, such as PP1 or PP2A, the catalytic and regulatory domains of PP5 are encoded on a single polypeptide (Shi, 2009; Swingle et al, 2004). This results in a low basal activity for PP5 as it adopts an autoinhibited conformation where its extreme C-terminal αJ helix interacts with the TPR domain in the N-terminus and blocks substrate access to the catalytic site (Kang et al, 2001). Many mechanisms of PP5 activation have been described, including the interaction of Hsp90 or Rac1 with the TPR domain and binding of polyunsaturated fatty acids, which classically were thought to release the autoinhibition (Chatterjee et al, 2010; Chen and Cohen, 1997; Haslbeck et al, 2015; Yang et al, 2005; Zeke et al, 2005). We have also shown previously that post-translational modification of PP5 can affect its activity. Casein kinase 1δ (CK1δ)-mediated phosphorylation of PP5-T362 leads to hyperactivity and supports cell survival in clear cell renal cell carcinoma (Dushukyan et al, 2017).

Recent advancements in the determination of PP5 structure have contributed significantly to our understanding of the relationship between PP5 structure and function in vitro (Jaime-Garza et al, 2023; Oberoi et al, 2016; Oberoi et al, 2022). Early structural work identified two essential metal ions as well as several key residues for PP5 catalytic activity, including H304, mutation of which renders PP5 catalytically dead (Swingle et al, 2004). The crystal structure of PP5 in complex with a peptide fragment of the co-chaperone substrate Cdc37 provided further insight into the important substrate contacting residues of PP5 (Oberoi et al, 2016). Additionally, structures of PP5 in complex with Hsp90, Cdc37, and kinase clients BRAF and CRAF were recently reported (Jaime-Garza et al, 2023; Oberoi et al, 2022). PP5 can be seen in both autoinhibited and open, active conformations and switches between the two TPR-binding sites of the Hsp90 dimer to allow access to various sites within the substrate client (Oberoi et al, 2022). These new structural insights provide a better understanding of PP5 function and the relationship of conformation to its function in vitro (Jaime-Garza et al, 2023; Vaughan et al, 2008).

There are, however, many outstanding questions with regard to PP5 regulation and function, particularly within a cellular context. Previously, we have shown the role of T362 phosphorylation on

[1]Department of Urology, SUNY Upstate Medical University, 750 E. Adams St., Syracuse, NY 13210, USA. [2]Upstate Cancer Center, SUNY Upstate Medical University, 750 E. Adams St., Syracuse, NY 13210, USA. [3]Department of Biochemistry and Molecular Biology, SUNY Upstate Medical University, 750 E. Adams St., Syracuse, NY 13210, USA. [4]School of Cancer and Pharmaceutical Sciences, Institute of Pharmaceutical Science, King's College London, London SE1 9NQ, UK. [5]These authors contributed equally: Rebecca A Sager, Sarah J Backe. ✉E-mail: mollapom@upstate.edu

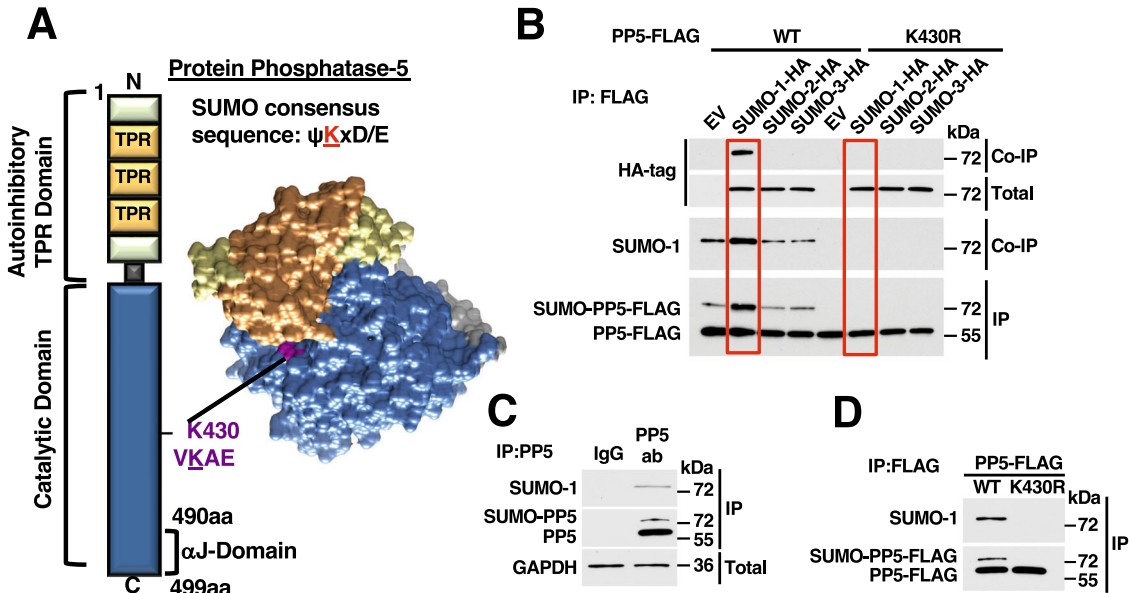

**Figure 1. SUMOylation of K430 in the catalytic domain of PP5.**

(A) Domain structure of PP5 containing the N-terminal autoinhibitory (green), TPR-containing domain (orange) and C-terminal catalytic domain (blue). Extreme C-terminal αJ helix (aa 490–499) is responsible for autoinhibition through contact with the TPR domain. PP5-K430 within the catalytic domain highlighted in purple in PP5 structure (PDB: 1WAO; modeled with Chimera) as K430 fits within the consensus sequence for SUMOylation. (B) Wild-type PP5-FLAG or PP5-K430R were transiently co-expressed with empty vector (EV), SUMO-1-HA, SUMO-2-HA, or SUMO-3-HA. PP5-FLAG was immunoprecipitated (IP) and co-immunoprecipitation of SUMO-HA was examined by Western blotting. (C) Endogenous IP of PP5 from HEK293 cells demonstrates the presence of SUMO-PP5. IgG was used as a control. (D) PP5-FLAG WT and K430R were transiently expressed and IP. The presence of SUMO-1-conjugated PP5 was examined by immunoblot. Source data are available online for this figure.

PP5 activity; however, whether PP5 is subject to other PTMs and how they work collectively to regulate the dephosphorylation and release of substrates from PP5 remains unknown (Dushukyan et al, 2017). Additionally, while there are numerous identified PP5 substrates, there has not yet been a comprehensive analysis to identify PP5 substrates or generate a consensus motif for PP5 dephosphorylation.

Here, we demonstrate that PP5 is subject to SUMOylation on K430. SUMOylation is a reversible PTM which results from the conjugation of a small ubiquitin-like modifier (SUMO) protein to a lysine residue (Hickey et al, 2012). PP5 SUMOylation is necessary for dephosphorylation and release of substrates in cells. Mutation of PP5-K430 to non-SUMOylatable arginine results in the accumulation of phosphorylated substrate bound to PP5. Further, we generated both an interactome of the PP5-K430R mutant by mass spectrometry and a candidate consensus motif for PP5 substrates and utilized these tools to identify new possible candidate PP5 substrates.

## Results and discussion

### Identification of a SUMOylation site in the catalytic domain of PP5

We have previously shown that PP5 is modified by both phosphorylation and ubiquitination, which modulate its activity and stability, respectively (Dushukyan et al, 2017). The question remained, however, whether PP5 is subject to additional modifications and how

these cooperate to regulate PP5 function. SUMOylation remains a relatively understudied PTM, however, it has been demonstrated to play a key role in Hsp90 molecular chaperone function as well as in cellular processes such as cell cycle progression and protein trafficking and conditions including cancer and neurodegenerative diseases, among others (Eifler and Vertegaal, 2015; Mollapour et al, 2014; Yang et al, 2015). Examination of the amino acid sequence of PP5 identified one lysine residue, K430, in the catalytic domain that fit the consensus sequence for SUMOylation (Fig. 1A). There are three isoforms of SUMO (SUMO-1, SUMO-2, and SUMO-3) that can be conjugated to target proteins (Hickey et al, 2012). We first overexpressed HA-tagged constructs of each isoform in the presence of either PP5-FLAG-WT or PP5-FLAG-K430R, in which the candidate SUMO site of PP5 had been mutated to a non-SUMOylatable arginine residue. First, we found that PP5 is SUMOylated (Fig. 1B); only SUMO-1-HA co-immunoprecipitated (co-IP) with PP5-FLAG and this interaction was abrogated in the K430R mutant, suggesting that SUMO-1 is the only SUMO isoform that modifies PP5 (Fig. 1B). Of note, the SUMOylated form of PP5 migrates slightly slower on SDS-PAGE as a result of addition of the SUMO protein to PP5. SUMOylation of PP5 with endogenous SUMO-1 was then confirmed on an endogenous PP5 immunoprecipitation (IP) (Fig. 1C) as well as abrogation with the PP5-FLAG-K430R mutant (Fig. 1D). Here, we provided evidence that PP5 is subject to SUMOylation on K430.

### SUMOylation controls PP5 activity

Next, we wanted to understand the effect of K430-SUMOylation on PP5 activity. We first examined the effect of SUMOylation on PP5

activity in vitro. PP5-His$_6$-WT and the K430R mutant were expressed and purified from bacteria for use in an in vitro PP5-specific phosphatase assay. For this assay, custom synthesized phospho-S211-glucocorticoid receptor (GR) peptide was used as a specific substrate (Thermo Fisher Scientific), and PP5 activity was measured by assessing free phosphate release as a result of PP5-mediated peptide dephosphorylation. Both PP5-His$_6$-WT and the K430R mutant exhibited similar in vitro activity levels (Vmax PP5-WT = 16.6 nmol/sec and PP5-K430R 21.6 nmol/sec) (Figs. 2A and EV1A). We then confirmed the effect of the mutation of lysine to arginine in the PP5-K430R mutant in cells. PP5-FLAG-WT and PP5-K430R were transiently expressed in HEK293 cells, and dephosphorylation of the well-described PP5 substrate phosphorylated GR-S211 was examined as a surrogate for PP5 activity (Wang et al, 2007). Overexpression of PP5-WT but not PP5-K430R led to a decrease in GR-S211 phosphorylation, suggesting SUMOylation of PP5 contributes to its activity in vivo (Fig. 2B). Furthermore, overexpression of SUMO-1-HA further enhanced the activity of PP5-WT, leading to a further decrease in GR-S211 phosphorylation. GR phosphorylation following overexpression of both PP5-K430R and SUMO-1-HA, however, remained unchanged (Fig. 2B). Collectively, our data suggest that in the absence of any PTMs PP5-K430R is capable of dephosphorylating peptide substrate and the activity effects seen in vivo are a result of its inability to be SUMOylated and not a deleterious effect of the mutation.

To examine whether PP5-K430R retains the ability to bind to its substrates in vivo, we expressed and immunoprecipitated PP5-FLAG-WT and the K430R mutant from HEK293 cells. Our data showed that, in fact, PP5-K430R had markedly increased binding to known PP5 substrates such as GR, Cdc37, and FNIP1 (Fig. 2C) (Sager et al, 2019; Vaughan et al, 2008; Wang et al, 2007). We also obtained similar results with both K430A and K430Q mutants (Fig. EV1B). The phosphorylated forms of GR and Cdc37 could also be seen bound to PP5-K430R, further supporting the importance of PP5 SUMOylation for its activity in cells (Figs. 2C and EV1B). The increased binding of substrates to the K430R mutant, however, is unlikely to be due solely to the decreased phosphatase activity because no change in substrate binding was seen with the catalytically dead PP5-H304Q mutant (Fig. EV1C) (Swingle et al, 2004).

The vast majority of PP5 substrates are co-chaperones or clients of the Hsp90 molecular chaperone, and it is generally accepted that PP5 binding to Hsp90 facilitates PP5 activation and substrate engagement (Chen et al, 1996; Davies and Sanchez, 2005; Silverstein et al, 1997; Yang et al, 2005). Interestingly, the binding of the K430R mutant to the molecular chaperone Hsp90 was not increased despite the increased binding of PP5-FLAG-K430R to its substrates (Figs. 2C and EV1B). We, therefore, next sought to determine whether the increased interaction of PP5-K430R with its substrates was occurring independent of interaction with Hsp90. We mutated two residues (K97E/R101E) in the PP5 TPR domain that mediate binding to the MEEVD motif on the Hsp90 carboxy-domain in the context of the non-SUMO-K430R mutant (PP5-FLAG-K97E/R101E-K430R) (Yang et al, 2005). As expected, PP5-FLAG-K97E/R101E and PP5-FLAG-K97E/R101E-K430R did not interact with Hsp90, additionally, interaction with the substrate GR was also abrogated for both mutants (Fig. 2D). This suggested that PP5 interaction with Hsp90 was prerequisite for substrate binding and SUMOylation-mediated regulation of PP5 function.

To continue dissecting the role of SUMOylation on PP5 activity we created a constitutively active PP5 by releasing its autoinhibition and examined the impact on its PTM status. This was accomplished through the deletion of the extreme C-terminal αJ helix of PP5, which is responsible for maintaining PP5 autoinhibition through contact with the TPR domain (Kang et al, 2001; Yang et al, 2005). PP5-ΔαJ and PP5-K430R-ΔαJ were both active and capable of dephosphorylating GR-pS211 in cells (Fig. 2E). Of note, the PP5-ΔαJ mutant did still rely on the ability to contact Hsp90 through PP5-K97 and -R101 in order to dephosphorylate GR (Fig. EV1D). We then further examined whether the αJ deletion affected the SUMOylation of PP5. Interestingly, PP5-FLAG-ΔαJ was not SUMOylated (Fig. 2F), which suggested that there was potentially a priming signal needed prior to SUMOylation, and this had been bypassed through the αJ deletion. In fact, when we examined the ΔαJ for the presence of other PTMs we found that threonine phosphorylation was also absent, but, PP5-FLAG-K430R had elevated levels of pThr (Fig. 2F). Taken together, our data demonstrated that SUMOylation regulates PP5 activity and revealed a potential cross-talk between threonine phosphorylation and SUMOylation of PP5.

## Cross-talk of SUMOylation and phosphorylation regulates PP5 phosphatase activity

While there are reports of phosphorylation-dependent SUMOylation of various proteins, there is still relatively little known about the cross-talk of SUMOylation with other PTMs (Liu et al, 2021; Su et al, 2012a; Su et al, 2012b; Vu et al, 2007). Previously, we have shown that PP5-T362 is phosphorylated by the CK1δ kinase, and this phosphorylation event mediates the hyperactivity of PP5 (Dushukyan et al, 2017). Therefore, we set out to examine the cross-talk between T362 phosphorylation and K430-SUMOylation (Fig. 3A). Consistent with our earlier observation, we found that the non-SUMOylatable K430R mutant was hyper-phosphorylated on threonine residues compared to PP5-FLAG-WT (Figs. 2F and 3B). Interestingly, the non-phosphorylatable T362A mutant was not SUMOylated, while the phosphomimetic T362E was hyper-SUMOylated compared to the PP5-WT (Fig. 3B,C). Similarly, overexpression of CK1δ kinase led to an increase in PP5 SUMOylation in addition to the expected increase in threonine phosphorylation (Fig. 3D). Collectively, these data suggest that PP5-T362 phosphorylation is a required priming signal for SUMOylation of PP5-K430.

Phosphorylation of T362 is essential for PP5 activity in vivo (Dushukyan et al, 2017), therefore we next examined how the cross-talk of phosphorylation and SUMOylation affect PP5 activity. Our data showed that the phosphomimetic and non-SUMOylatable PP5 (T362E-K430R) double mutant, like the K430R mutant, was unable to dephosphorylate the PP5 substrate GR-pS211 in cells (Fig. 3E). Similarly, overexpression of CK1δ, the kinase that mediates PP5-T362 phosphorylation, enhanced the activity of PP5-FLAG-WT but not PP5-K430R (Figs. 3F and EV2A). These data suggest that threonine phosphorylation alone is insufficient for the phosphatase activity of PP5 in cells. Furthermore, our data suggested that PP5-T362 phosphorylation may commit a PP5-substrate complex to substrate dephosphorylation and eventual substrate release. The PP5-T362E mutant exhibited increased binding to its substrates (Fig. EV2B). Overexpression of CK1δ enhanced the activity of PP5-WT and increased its binding to GR

 

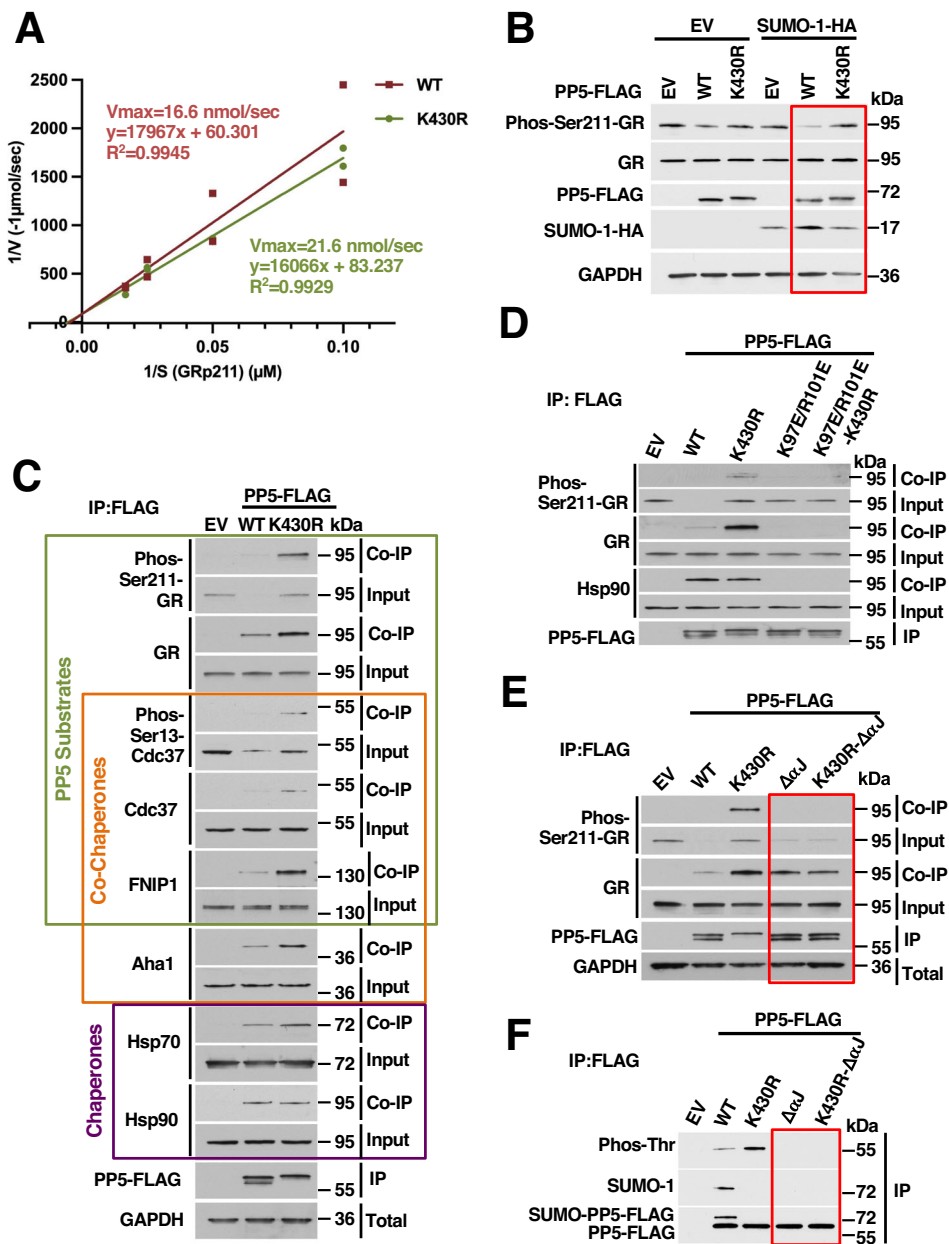

**Figure 2. SUMOylation controls PP5 activity.**

(A) PP5-His$_6$-WT and PP5-His$_6$-K430R were expressed and purified from bacteria. PP5 activity was examined in vitro using GR-pS211 phospho-peptide as a substrate and phosphate release measured using PiPer phosphate assay. Kinetics of WT and PP5-K430R activity represented using Lineweaver–Burk plot. The data shown are two technical replicates with a simple linear regression plot. (B) Wild-type PP5-FLAG or K430R mutant was transiently expressed with EV or SUMO-1-HA overexpressed. Activity of PP5 in vivo assessed through immunoblotting for phosphorylated GR-S211 level. EV was used as a control. GAPDH was used as a loading control. (C) PP5-FLAG WT and K430R were expressed and immunoprecipitated. Co-IP of known PP5 substrates, Hsp90 co-chaperones, and chaperones were examined by Western blot. (D) PP5-FLAG WT, K430R, and TPR domain mutants PP5-K97E/R101E and PP5-K97E/R101E-K430R were transiently expressed and immunoprecipitated. The binding of Hsp90 and the substrate GR were examined by immunoblot. EV was used as a control. (E) PP5-FLAG WT, K430R, the activated PP5-ΔαJ, and double PP5-K430R-ΔαJ mutants were transiently expressed and immunoprecipitated. The Binding of the substrate GR were examined by immunoblot. EV was used as a control. (F) Wild-type PP5 and the mutants used in (E) were transiently expressed. PP5-FLAG was isolated by IP and examined for PTMs, including SUMO and threonine phosphorylation. EV was used as a control. Source data are available online for this figure.

while treatment with the CK1δ inhibitor IC261 decreased PP5 activity and slightly decreased PP5-WT interaction with its substrate GR (Fig. EV2C–E). Lastly, the enhanced co-IP of GR with phosphomimetic PP5-FLAG-T362E was similar to that seen

with the K430R mutant as well as the T362E-K430R double mutant (Fig. EV2F). Our data demonstrate that PP5-T362 phosphorylation primes PP5 for subsequent SUMOylation and commits the PP5:substrate complex to substrate dephosphorylation and release

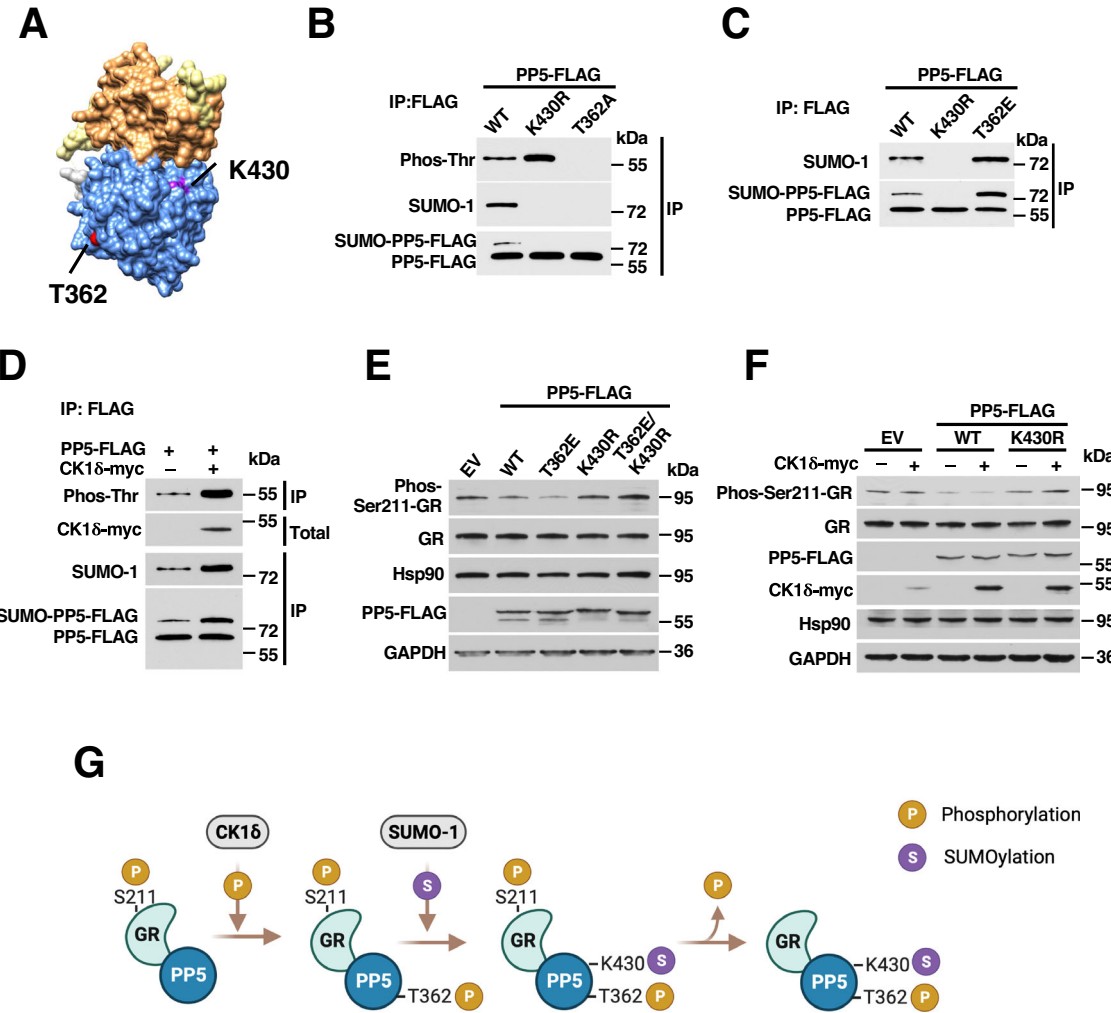

**Figure 3.  Cross-talk of SUMOylation and phosphorylation regulates PP5 phosphatase activity.**

(A) Space-filled PP5 crystal structure with T362 (red) and K430 (purple) highlighted within the catalytic domain (blue) (PDB: 1WAO; modeled with Chimera). (B) Wild-type PP5-FLAG, non-SUMOylatable K430R, and non-phosphorylatable T362A mutants were expressed and isolated by IP. PP5-FLAG PTMs, including SUMOylation and threonine phosphorylation, were examined by Western blot. (C) Wild-type PP5-FLAG, non-SUMOylatable K430R, and phosphomimetic T362E mutants were expressed and isolated by IP. PP5-FLAG SUMOylation was examined by Western blot. (D) PP5-FLAG was transiently transfected with EV or CK1δ-myc. Following the IP of PP5-FLAG, threonine phosphorylation and SUMOylation of PP5-FLAG was examined by Western blot. (E) PP5-FLAG WT, phosphomimetic T362E, non-SUMOylatable PP5-K430R, or the PP5-T362E-K430R double mutant were transiently transfected. The activity of PP5-FLAG towards dephosphorylation of total GR-S211 phosphorylation was assessed by immunoblotting. EV was used as a control. (F) PP5-FLAG WT or K430R were co-transfected with EV or CK1δ-myc. The activity of PP5-FLAG toward dephosphorylation of total GR-S211 phosphorylation was assessed by immunoblotting. EV was used as a control. (G) Schematic model of ordered addition of PP5 PTMs where phosphorylation of PP5-T362 is prerequisite to SUMOylation of PP5-K430. Both phosphorylation and SUMOylation of PP5 are needed for its ability to dephosphorylate its substrate, GR. Source data are available online for this figure.

(Fig. 3G). This ordered sequence of PTMs is ultimately necessary for PP5 activity and substrate release in cells (Fig. 3G).

## PP5 SUMOylation is essential for the substrate release in cells

Our in vitro and in vivo data above suggested, and was consistent with previously published results, that binding of substrates such as GR to PP5 is achieved via Hsp90 delivery through TPR-domain mediated contact into an "early" PP5:Hsp90:substrate complex. Our data also suggests that SUMOylation of PP5-K430 is essential for progression through the late complex, substrate

dephosphorylation, and full substrate release (Fig. 3G). However, it is unclear whether there is a step at which the assistance of Hsp90 is no longer required. In order to dissect these later stages of PP5-mediated dephosphorylation of its substrates, we first isolated PP5-GR complexes from HEK293 cells and showed that Hsp90 was present in this complex (Fig. 4A). We also utilized the K430R mutant that increased binding to the substrates but not Hsp90 and demonstrated that Hsp90 was also present in this complex (Fig. 4A). Next, we questioned whether the phosphorylated form of GR, which serves as substrate for PP5, remained in complex with Hsp90. Endogenous phospho-S211-GR was immunoprecipitated from HEK293 cells. Co-immunoprecipitation of both PP5 and

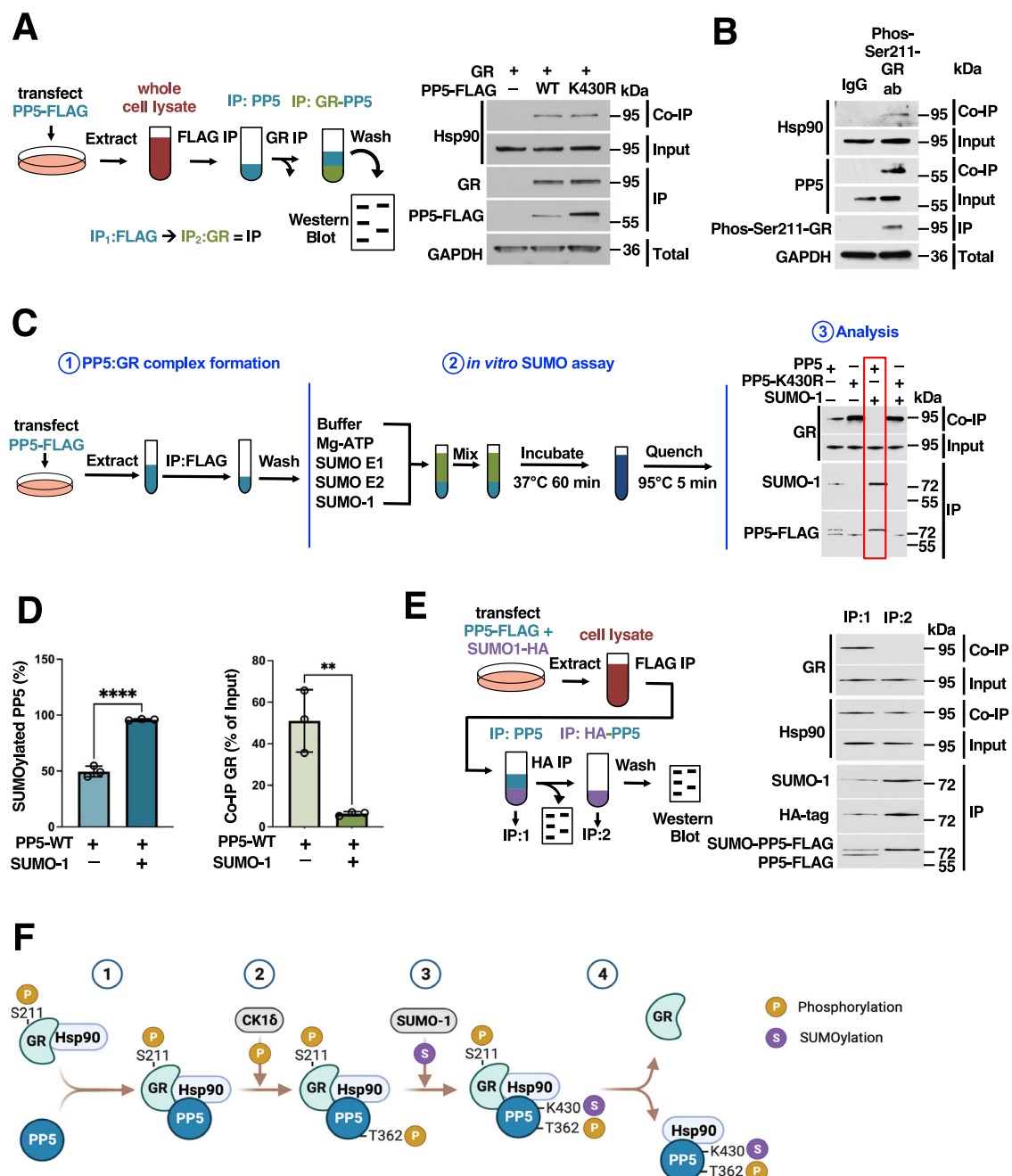

Hsp90 with the phosphorylated form of GR was visible (Fig. 4B). Together, this suggests that a complex likely exists containing Hsp90, PP5, and phosphorylated GR. We then asked if PP5 SUMOylation contributes toward substrate release. PP5-WT and K430R were isolated from HEK293 cells and used together with SUMO-1 in an in vitro SUMOylation assay (Enzo Life Sciences). Our data showed hyper-SUMOylation of PP5-WT with SUMO-1 and a complete lack of SUMOylation of PP5-K430R (Figs. 4C and EV3). We assessed substrate release by evaluating GR binding to PP5 in the presence of SUMO-1. Notably, we were unable to detect GR bound to hyper-SUMOylated PP5-WT (Figs. 4C,D and EV3), whereas lack of SUMOylation in the K430R mutant led to

enhanced interaction of GR with PP5 (Figs. 4C and EV3). This data suggests that SUMOylation of PP5 promotes substrate release. Notably, as the PP5:GR complex was not bacterially expressed and purified, there may be additional components present that contribute to this process. Therefore, we next sought to confirm the impact of SUMOylation on substrate release in vivo. We have already shown that overexpression of SUMO-1-HA enhanced the activity of PP5-WT but not PP5-K430R toward the dephosphorylation of GR-S211 (Fig. 2B). Here, we co-overexpressed PP5-FLAG-WT and SUMO-1-HA in HEK293 cells and then isolated SUMO-1-PP5 by sequential IPs (FLAG IP followed by HA IP). Our data confirm the inability of SUMO-PP5 binding to the

**Figure 4. PP5 SUMOylation is essential for the substrate release in cells.**

(A) EV, PP5-FLAG WT, or PP5-K430R were transiently transfected with GR. PP5-FLAG was immunoprecipitated from cell lysate. Following FLAG IP₁, peptide competition was performed with FLAG peptide for elution, and a second round of IP₂ was performed using GR antibody-conjugated agarose. Following the second IP₂, isolated protein complexes were eluted and analyzed by immunoblotting. (B) Phosphorylated GR-pS211 was immunoprecipitated from HEK293 cell lysates using anti-GR-pS211 or IgG as control. Interacting proteins were analyzed by immunoblot. (C) PP5-FLAG WT or K430R were expressed and isolated for use in an in vitro SUMOylation assay. PP5-WT and K430R were incubated with ATP in the presence and absence of recombinant SUMO-1. FLAG IP was used to isolate PP5 following in vitro SUMOylation and interaction with GR was evaluated by immunoblot. PP5 SUMOylation was assessed by immunoblotting. Data shown is the same as replicate three in Fig. EV3. (D) Densitometric analysis of Fig. 4C. Left graph represents the percentage of PP5-WT that is SUMOylated without or with in vitro SUMOylation by SUMO-1. The right graph represents the Co-IP of GR with WT-PP5 (normalized to GR input) without or with in vitro SUMOylation by SUMO-1. Data were presented as mean ± standard deviation derived from three biologiclal replicates ($n = 3$). An unpaired $t$-test was used to determine statistical significance. ****$p = 0.000083$, **$p = 0.0068$. (E) PP5-FLAG WT was co-transfected with SUMO-1-HA in HEK293 cells. FLAG IP was used to isolate PP5-FLAG (SUMOylated and non-SUMOylated) (IP:1). A subsequent HA IP was used to isolate only SUMOylated PP5 (IP:2). IP:1 and IP:2 were analyzed for interaction with Hsp90 and GR by immunoblot. (F) Schematic model demonstrating the proposed impact of PP5 PTMs on the regulation of GR activity through GR phosphorylation. GR interacts with its chaperone Hsp90, and the interaction of PP5 with Hsp90 via PP5-K97/R101 is necessary to mediate the interaction of GR with PP5 (1). The phosphorylated form of GR, GR-pS211, can be seen in complex with both Hsp90 and PP5. PP5 requires sequential phosphorylation on T362 (2) and subsequent SUMOylation on PP5-K430 (3) for PP5 activity and the ability to dephosphorylate and release its substrate GR (4). Source data are available online for this figure.

GR substrate (Fig. 4E). Interestingly, SUMOylation of PP5 did not impact its binding to Hsp90, suggesting that PP5 SUMOylation specifically releases the substrate from the PP5 complex (Fig. 4E). Collectively, our data supports a model where Hsp90 mediates the initial interaction of PP5 and GR. Through the ordered phosphorylation of PP5-T362 and subsequent SUMOylation of K430, PP5 is able to dephosphorylate and release its substrate GR (Fig. 4F). However, SUMOylation does not affect PP5 interaction with Hsp90 and, therefore, this PTM specifically releases substrates from the PP5 complex in a cellular context (Fig. 4F).

## Identification of novel PP5 substrate candidates

While numerous PP5 substrates have been described and reviewed previously there was not yet a comprehensive analysis or identification of a motif for PP5 substrate dephosphorylation. This prompted us to generate a candidate dephosphorylation consensus motif based on known PP5 substrate residues (Appendix Table S1), (Amable et al, 2011; Hinds and Sanchez 2008; Hu et al, 2018; Ikeda et al, 2004; Kang et al, 2009; Krysiak et al, 2018; Liu et al, 2002; Mazalouskas et al, 2014; von Kriegsheim et al, 2006; Wang et al, 2009; Wechsler et al, 2004). We found that the target serine or threonine is often followed by a proline residue and/or is preceded by an acidic residue, either aspartate or glutamate (Fig. 5A). As the PP5-K430R mutant bound more to known substrates–including GR, Cdc37, and FNIP1– relative to PP5-WT (Fig. 2C), we utilized the PP5-K430R mutant as a substrate "trap" for proteomic analysis with the goal of identifying novel PP5 substrate candidates. Both PP5-FLAG-WT and PP5-K430R were transiently transfected in HEK293 cells and immunoprecipitated (Fig. EV4A). Co-immunoprecipitated proteins were then subject to analysis by mass spectrometry (Fig. 5B; Dataset EV1). We identified 87 proteins bound to PP5-K430R in significantly higher abundance than the WT-PP5, and they were involved in cellular processes such as signaling and post-translational modifications, RNA processing and transcription, and cell cycle regulation, among others (Fig. 5B). We next used our proposed consensus motif to further analyze our interactome with the goal of confirming candidate PP5 substrates. Approximately 90% of the proteins identified by mass spectrometry as bound more to PP5-K430R contained at least one S/T-P motif (Fig. 5C). Further analysis identified 35% of the proteins which bound more to the K430R mutant than PP5-WT contained the

more stringent E/D-S/T-P motifs and 42% of them contained either E/D-S/T-P or pS/pT/pY-S/T-P (Fig. 5C). This is supported by prior work that found a high abundance of proline-directed kinase phosphorylation sites as potential PP5 substrates in response to DNA damage (Ham et al, 2010). This study also found enrichment in casein kinase-2 (CK2) phosphorylation sites for possible PP5 targets (Ham et al, 2010). This has been confirmed with two substrates of PP5, including Cdc37 and FNIP1 (Miyata, 2005; Sager et al, 2019; Vaughan et al, 2008).

To confirm the importance of the residues at the −1 and +1 positions in our dephosphorylation motif, we generated multiple mutants of GR, by mutating the E210 and P212 surrounding the S211 residue, both of which fit within our proposed motif, to alanine. These were expressed in HEK293 cells alone and with PP5 co-expressed. We found that the single GR-E210A and GR-P212A mutants were markedly less phosphorylated on GR-S211 (Fig. EV4B). The overexpression of PP5, however, was still able to decrease this phosphorylation further, although to a lesser extent than the GR-WT (Fig. EV4B). Phosphorylation on S211 of the double mutant GR-E210A/P212A, however, was not visible regardless of whether PP5 was overexpressed, therefore we were unable to evaluate PP5-mediated dephosphorylation in this context (Fig. EV4C). Taken together, we believe this motif likely helps to drive some of the specificity of PP5-mediated dephosphorylation towards these substrates. It is likely that additional substrate specificity is driven by an interaction motif separate from the dephosphorylation motif, such as the generalized TPR-binding motif (EEVD) that was recently identified (Devi et al, 2024). This EEVD-like binding motif in combination with our proposed dephosphorylation motif, may help in the identification of new candidate substrate residues in other known interactors.

It is notable that the majority of known PP5 substrate residues are serine and therefore suggests PP5 may prefer serine over threonine (Appendix Table S1). We used a commercially available phospho-specific antibody for one of the new candidate substrates fitting the pS-P motif, DNA topoisomerase II alpha-S1213 (TOP2A-S1213). Our data showed that phosphorylation of TOP2A-S1213 decreased following overexpression of WT-PP5-FLAG but not the PP5-K430R mutant (Fig. 5D). Notably, there are 7994 genes in the human genome containing the E/D-S-P motif, (8094 contain the less stringent S-P motif) (RSCB.org) (Berman et al, 2000), while the PhosphoSitePlus database (Phosphosite.org,

 

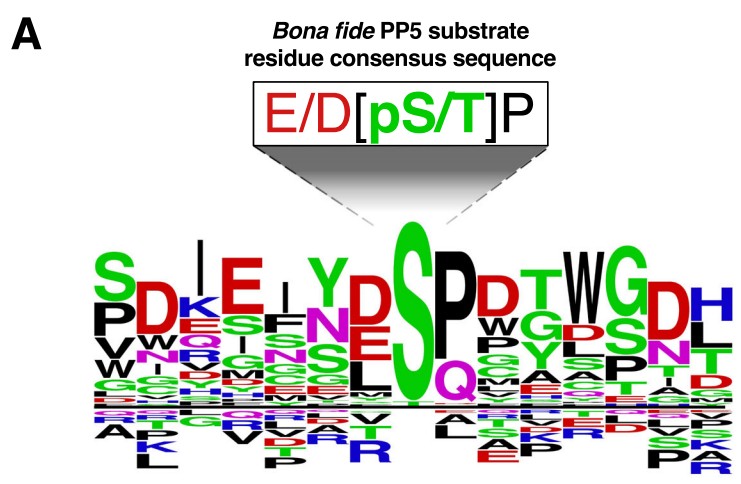

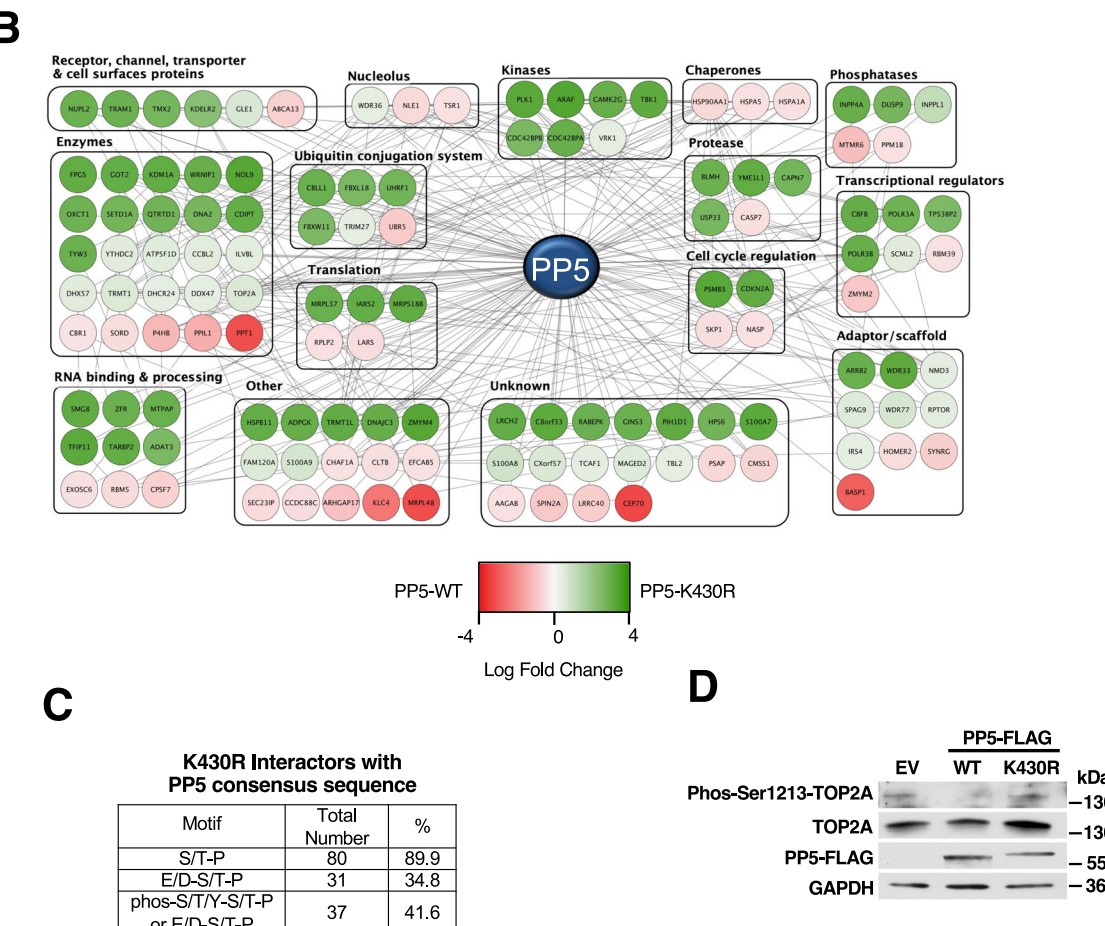

## C

**K430R Interactors with PP5 consensus sequence**

| Motif | Total Number | % |
|---|---|---|
| S/T-P | 80 | 89.9 |
| E/D-S/T-P | 31 | 34.8 |
| phos-S/T/Y-S/T-P or E/D-S/T-P | 37 | 41.6 |

## D

(Hornbeck et al, 2015) reports only 2130 proteins with phosphorylation of the serine within E/D-S-P (7780 within S-P) (Appendix Table S2). It remains unclear which residues are the most important in the surrounding sequence, though the acidic residues at −1, proline at +1, and glycine at +5 are quite prominent in the motif. Addressing this question is challenging because mutations within this motif may abrogate the phosphorylation of the S/T residues that are targeted by PP5, such as we have shown for GR (Fig. EV4B,C).

Taken together, interactome analysis of the PP5-K430R mutant provided identification of novel PP5 substrate candidates in conjunction with our working consensus motif, which can be used

**Figure 5. Identification of novel PP5 substrate candidates.**

(**A**) The candidate consensus motif for PP5 dephosphorylation was determined using known PP5 substrate residues as inputs in PhosphoSitePlus v.6.6.0.4 Sequence Logo tool. (**B**) PP5-FLAG WT or K430R was transiently transfected and immunoprecipitated from HEK293 cells. PP5-FLAG and interacting proteins bound to anti-FLAG conjugated agarose was digested and analyzed by mass spectrometry to identify interacting proteins. Nodes represent proteins identified by mass spectrometry; lines represent reported interactions in string.db. Proteins were grouped based on "protein type" identifier in PhosphoSitePlus database and color-coded based on the log fold change of intensity in the K430R sample over WT. (**C**) Amino acid sequences of proteins identified by mass spectrometry to interact more with non-SUMOylatable PP5-K430R ("K430R Interactors"; PP5-K430R/WT fold change >2) were manually searched for the presence of each motif. The table shows the total number of K430R interactors found to contain the motif, and the percentage of K430R interators containing at least one motif. (**D**) PP5-FLAG WT or K430R were transiently overexpressed in HEK293 cells. Phosphorylation of candidate substrate TOP2A-S1213 and total TOP2A expression were analyzed by immunoblot. EV was used as a control. Source data are available online for this figure.

for future studies. Collectively, this study further unravels the mechanisms governing PP5 regulation by PTMs within cells. We have built upon prior identified modifications to understand how these and our newly identified SUMOylation site cooperate to regulate PP5 function. Additionally, we have gained insight into the functional regulation of PP5 phosphatase within a cellular environment. This information will help guide future studies to understand how this is altered in cancers that depend on PP5 activity and design a strategy to target and inhibit PP5.

# Methods

### Reagents and tools table

| Reagent/resource | Reference or source | Identifier or catalog number |
|---|---|---|
| **Experimental models** | | |
| HEK293 | ATCC | Cat# CRL-1573 |
| **Recombinant DNA** | | |
| pcDNA3-PP5-FLAG | (Dushukyan et al, 2017) | n/a |
| pcDNA3-PP5-FLAG-T362A | (Dushukyan et al, 2017) | n/a |
| pcDNA3-PP5-FLAG-T362E | (Dushukyan et al, 2017) | n/a |
| pcDNA3-PP5-FLAG-H304Q | (Ahanin et al, 2023) | n/a |
| pcDNA3-SUMO-1-HA | (Mollapour et al, 2014) | n/a |
| pcDNA3-SUMO-2-HA | (Mollapour et al, 2014) | n/a |
| pcDNA3-SUMO-3-HA | (Mollapour et al, 2014) | n/a |
| pcDNA3-CK1δ-myc | (Dushukyan et al, 2017) | n/a |
| GR-FLAG | (Backe et al, 2022) | n/a |
| pRSETA-PP5-His$_6$ | (Dushukyan et al, 2017) | n/a |
| **Antibodies** | | |
| Rabbit anti-FLAG tag | Thermo Scientific | Cat# PA1-984B; RRID:AB_347227 |
| Mouse anti-FLAG tag | Thermo Scientific | Cat# F3165; RRID:AB_259529 |

| Reagent/resource | Reference or source | Identifier or catalog number |
|---|---|---|
| Mouse anti-6x-His epitope tag (HIS.H8) | Thermo Scientific | Cat# MA1-21315; RRID:AB_557403 |
| Rat anti-Hsp90 (16F1) | Enzo Life Sciences | Cat# ADI-SPA-835; RRID:AB_11181205 |
| Mouse anti-Hsp90 (F-8) | Santa Cruz Biotechnology | Cat# sc-13119; RRID:AB_675659 |
| Mouse anti-GAPDH (1D4) | Enzo Life Sciences | Cat# ADI-CSA-335; RRID:AB_10617247 |
| Mouse anti-SUMO-1 (2A12) | Cell Signaling Technology | Cat# 5718; RRID:AB_10547145 |
| Rabbit anti-PP5 | Cell Signaling Technology | Cat# 2289; RRID:AB_2168757 |
| Mouse anti-PP5 (2E12) | Abcam | Cat# ab123919; RRID:AB_10976136 |
| Rabbit anti-phos-Ser13-Cdc37 (EPR4979) | Abcam | Cat# ab108360; RRID:AB_10859480 |
| Rabbit anti-Cdc37 | StressMarq Biosciences | Cat# SPC-142; RRID:AB_2570605 |
| Rabbit anti-Aha1 | StressMarq Biosciences | Cat# SPC-183; RRID:AB_1944037 |
| Rabbit anti-Hsp70 | StressMarq Biosciences | Cat# SPC-103 RRID:AB_2570584 |
| Rabbit anti-GR (D6H2L) | Cell Signaling Technology | Cat# 12041; RRID:AB_2631286 |
| Mouse anti-GR (D4X9S) | Cell Signaling Technology | Cat# 47411; RRID:AB_2799324 |
| Rabbit anti-phospho-GR-S211 | Cell Signaling Technology | Cat# 4161; RRID:AB_2155797 |
| Rabbit anti-HA tag (C29F4) | Cell Signaling Technology | Cat# 3724; RRID:AB_1549585 |
| Rabbit anti-myc tag (71D10) | Cell Signaling Technology | Cat# 2278; RRID:AB_490778 |
| Mouse anti-phospho-threonine (PTR-8) | Sigma-Aldrich | Cat# P6623; RRID:AB_477393 |
| Rabbit anti-phos-Ser1213-TOP2A | Invitrogen | Cat# PA5-105828; RRID:AB_2817227 |
| Rabbit anti-TOP2A (D10G9) | Cell Signaling Technology | Cat# 12286; RRID:AB_2797871 |
| Goat anti-FNIP1 | Antibodies-online.com | Cat# ABIN238670; RRID:AB_10775640 |
| Anti-mouse secondary | Cell Signaling Technology | Cat# 7076; RRID:AB_330924 |

 

| Reagent/resource | Reference or source | Identifier or catalog number |
|---|---|---|
| Anti-rabbit secondary | Cell Signaling Technology | Cat# 7074; RRID:AB_2099233 |
| Anti-rat secondary | Cell Signaling Technology | Cat# 7077; RRID:AB_10694715 |
| Anti-goat secondary | Santa Cruz Biotech | Cat# sc-2020; RRID:AB_631728 |
| **Oligonucleotides and other sequence-based reagents** | | |
| DNA primers | Eurofins Genomics | Appendix Table S3 |
| **Chemicals, enzymes, and other reagents** | | |
| IC261 | Abcam | Cat# ab145189 |
| Phos-Ser211-GR peptide ([NH2]PGKETNE[pS]PWRSDLL[COOH]) | Thermo Fisher Scientific custom synthesized | This paper |
| 3x FLAG peptide | Sigma-Aldrich | Cat# F4799 |
| NEM (N-ethylmaleimide) | Sigma-Aldrich | Cat# E3876 |
| BL21(DE3) | EMD Millipore | Cat# 69450 |
| DH5-alpha Electrocompetent E coli | Goldbio | Cat# CC-203 |
| **Software** | | |
| Biorender | https://biorender.com/ | n/a |
| UCSF Chimera, candidate version 1.14 | (Pettersen et al, 2004) | https://www.cgl.ucsf.edu/chimera/ |
| Cytoscape 3.9.0 | (Gustavsen et al, 2019) | https://doi.org/10.12688/f1000research.20887.3 |
| GraphPad Prism version 9.2.0 for macOS | | GraphPad Software, La Jolla, California, USA, www.graphpad.com |
| PhosphoSitePlus v6.6.0.4 | (Hornbeck et al, 2015) | https://www.phosphosite.org/ |
| RCSB PDB | (Berman et al, 2000) | RCSB.org |
| **Other** | | |
| Mirus TransIT-2020 | MirusBio | Cat# MIR5405 |
| PiPer Phosphate Assay | Thermo Fisher Scientific | Cat# P22061 |
| Anti-FLAG M2 affinity gel | Sigma-Aldrich | Cat# A2220 |
| Protein G agarose | Thermo Fisher Scientific | Cat# 15-920-010 |
| GR (FiGR) antibody-conjugated agarose | Santa Cruz Biotechnology | Cat# sc-12763 |
| Ni-NTA Agarose | Thermo Fisher Scientific | Cat# 88221 |
| In vitro SUMOylation Kit | Enzo Life Sciences | Cat# BML-UW8955-0001 |

## Materials availability

Materials generated in this study will be made available on request, but we may require a payment and/or a completed materials transfer agreement if there is potential for commercial application.

## Cell lines

Cultured human embryonic kidney (HEK293) cells were grown in Dulbecco's Modified Eagle Medium (DMEM, Sigma-Aldrich) supplemented with 10% fetal bovine serum (FBS, Sigma-Aldrich). HEK293 cells were acquired from (the American Type Culture Collection, ATCC). Cells were maintained in a CellQ incubator (Panasonic Healthcare) at 37 °C in an atmosphere containing 5% $CO_2$. To verify that these are not false cell lines, misidentified, and are authentic stock, we will periodically check the Cellosaurus web pages as we understand that this resource houses the most up-to-date information on cell line misidentification and works closely with cell line repositories.

## Plasmids

For mammalian expression, pcDNA3-PP5-FLAG and the T362A and T362E point mutations, pRSETA-PP5-His$_6$ as well as pcDNA3-CK1δ-myc were created previously (Dushukyan et al, 2017). pcDNA3-PP5-FLAG-H304Q point mutant was reported previously (Ahanin et al, 2023). SUMO-1-HA, SUMO-2-HA, and SUMO-3-HA were also created previously (Mollapour et al, 2014). GR-FLAG expression plasmid was used previously (Backe et al, 2022). Point mutations were made using site-directed mutagenesis (Appendix Table S3) and confirmed by DNA sequencing.

## Cell transfection and drug treatment

Cultured HEK293 cells were split and then transfected the following day when about 40% confluent with plasmid DNA using Mirus TransIT-2020 (MirusBio) according to the manufacturer's protocol. Cells were extracted or treated with IC261 (Abcam) the following day.

## Protein extraction, immunoprecipitation, and immunoblotting

Protein extraction from mammalian cells was carried out using methods previously described with the addition of 20 mM N-ethylmaleimide (NEM) (Sigma-Aldrich) in the lysis buffer (Mollapour et al, 2010; Sager et al, 2018). For immunoprecipitation, mammalian cell lysates were incubated with anti-FLAG antibody-conjugated agarose beads (Sigma-Aldrich) for 2 h at 4 °C. Immunopellets were washed four times with fresh lysis buffer (20 mM Tris-HCl (pH 7.4), 100 mM NaCl, 1 mM MgCl$_2$, 0.1% NP40, protease inhibitor cocktail (Roche), and PhosSTOP (Roche)) and eluted in 5x Laemmli buffer. In order to visualize post-translational modifications, immunopellets were washed four times in lysis buffer with high salt (as above with 500 mM NaCl) prior to elution in 5x Laemmli buffer. In sequential double IP experiments, the first IP was competed with 3x FLAG peptide (Sigma-Aldrich) prior to the second IP as previously described (Woodford et al, 2017). The second IP for FLAG:GR double IP was performed using anti-GR conjugated agarose (Santa Cruz Biotechnology). Endogenous protein immunoprecipitation was performed by pre-clearing lysate with protein G agarose (Thermo Fisher Scientific) for 1 h, incubation overnight at 4 °C with primary antibody, followed by incubation with protein G agarose for 2 h. After washing, proteins

were eluted in 5x Laemmli buffer. Precipitated proteins were separated by SDS-PAGE and transferred to nitrocellulose membranes. Co-immunoprecipitated proteins or proteins from cell lysate were detected by immunoblotting with antibodies recognizing FLAG, 6x-His (HIS.H8) (Thermo Fisher Scientific), Hsp90 (16F1), GAPDH (ENZO Life Sciences), GR (D6H2L), Myc (71D10), SUMO-1 (2A12), HA (C29F4), phospho-S211-GR, PP5, GR (D4X9S), TOP2A (D10G9) (Cell Signaling Technology), phos-Ser1213-TOP2A (Invitrogen), FNIP1 (antibodies-online.com), PP5 (2E12), phospho-S13-Cdc37 (EPR4979) (Abcam), Aha1, Cdc37, Hsp70 (StressMarq Biosciences), Hsp90 (F-8) (Santa Cruz Biotechnology), or phospho-Thr (PTR-8) (Sigma-Aldrich). Secondary antibodies raised against mouse, rabbit, and rat (Cell Signaling Technology) and goat (Santa Cruz Biotechnology) were used. Western blot densitometry analysis was performed using ImageJ.

## Bacterial expression and protein purification

PP5-His$_6$ WT and K430R mutants were expressed and purified from *E. coli* strain BL21 (DE3) (EMD Millipore). Transformed cells were grown at 37 °C in LB with 50 mg/L ampicillin until $OD_{600} = 0.6$. The cultures were then cooled to 30 °C, induced with 1 mM IPTG, and grown to $OD_{600} = 1.2$. Cells were harvested by centrifugation and lysed by sonication in fresh lysis buffer without detergent (20 mM Tris-HCl (pH 7.4), 100 mM NaCl, 1 mM MgCl$_2$, protease inhibitor cocktail (Roche), and PhosSTOP (Roche)). The supernatant was collected, and PP5 expression was assessed by immunoblotting. Isolation of PP5-His$_6$ was accomplished by two sequential Ni-NTA agarose (Thermo Fisher Scientific) pulldowns. Lysate was incubated with Ni-NTA agarose for 2 h at 4 °C. Proteins bound to Ni-NTA agarose were washed three times with lysis buffer (as above) followed by two washes with 50 mM imidazole in lysis buffer. They were then eluted in 500 mM imidazole in lysis buffer and concentrated in 30 K Amicon® Ultra Centrifugal Filters (Millipore). Concentrations were determined using the Micro BCA™ Protein Assay Kit (Thermo Scientific) per manual protocol. Purified protein was run on an SDS-PAGE gel and Coomassie stained to confirm purity prior to use in assays.

## In vitro SUMOylation assay

SUMOylation kit (Enzo Life Sciences Inc) was used for the in vitro SUMOylation of PP5-FLAG WT and K430R mutant IPed from HEK293 cells. Reaction containing 1X SUMO activation enzyme, 1X SUMO conjugating enzyme, 1X SUMO-1 solution, 1X SUMOylation buffer, and 1XMg$^{2+}$-ATP buffer per manual protocol. The reaction was initiated at 37 °C for 60 min. The assay was quenched by the addition of 5x Laemmli buffer, followed by boiling. Samples were separated by SDS-PAGE and transferred to nitrocellulose membranes followed by Western blot analysis.

## PP5 phosphatase activity assay

PP5-His$_6$ and K430R mutant were expressed and purified from bacteria. The phosphatase activity of these proteins were measured using a using the PiPer™ Phosphate Assay Kit (Thermo Fisher Scientific) as described in the manufacturer's protocol. A standard curve with a linear fit line was created from 0–1 nM P$_i$ final concentration reactions. About 1 nM of PP5-His$_6$ or PP5-His$_6$-

K430R mutant were added to each reaction with indicated amounts of custom synthesized phospho-S211-glucocorticoid receptor (GR) peptide (Phos-Ser211-GR peptide ([NH2]PGKETNE[pS] PWRSDLL[COOH]) as specific substrate (Thermo Fisher Scientific). Reactions were run in duplicate and incubated at 37 °C for over 10 min. Enzyme kinetic was calculated and plotted using the Lineweaver–Burk plot. PP5-His$_6$ or PP5-His$_6$-K430R phosphatase activities were presented as Vmax.

## Mass spectrometry analysis

HEK293 cells were transfected with EV, pcDNA3-PP5-FLAG WT or PP5-FLAG-K430R. Protein lysate was extracted, and FLAG immunoprecipitation was performed as above. After the final washing of the FLAG agarose, agarose and bound proteins was stored at −80 °C. A small aliquot of each sample was eluted with 5x Laemmeli for Coomassie staining and Western blot analysis. The remainder was subjected to protease digestion and then loaded directly onto a 50 cm × 100 μM PicoFrit capillary column packed with C18 reversed-phase resin. The column was developed with a 150-min linear gradient of acetonitrile in 0.125% formic acid delivered at 280 nL/min. LC-MS/MS Analysis was done using Orbitrap-Fusion Lumos, ESI-HCD. Duplicate injections were performed for each sample. MS/MS spectra were evaluated using SEQUEST (Eng et al, 2008) and the Core platform from Harvard University. Searches were performed against the most recent update of the NCBI (refseq) Homo sapiens database with a mass accuracy of ±50 ppm for precursor ions and 0.02 Da for product ions. Results were filtered with mass accuracy of ±5 ppm on precursor ions and presence of the intended motif. Results were further filtered to a 1% protein false discovery rate. Data are presented in Dataset EV1.

The relative fold change between conditions was determined by the integrated peak area of the experimental (numerator) and control (denominator) conditions. Proteins with a fold change of 3 or greater over the control were selected for further analysis. Fold change was then calculated based on the integrated peak area for proteins in the PP5-K430R mutant over the PP5-WT sample. Proteins with an absolute value fold change >2 were uploaded to string.db. Nodes and connections were downloaded from string.db and imported into Cytoscape v3.9.0 for figure construction (Gustavsen et al, 2019). Proteins were color-coded by log(fold change), with green indicating increased abundance in the PP5-K430R sample compared to the PP5-WT sample. Red indicates proteins with a lower abundance in PP5-K430R than PP5-WT. Proteins were grouped based on "protein type" identifier in PhosphoSitePlus database (Hornbeck et al, 2015).

## Consensus sequence analysis

Known PP5 substrate residues (Appendix Table S1) were entered in PhosphoSitePlus v.6.6.0.4 Sequence Logo tool (Hornbeck et al, 2015). Phospho Ser/Thr background with PSP production settings were used to analyze the peptide (−7 to +7) surrounding the modified serine or threonine residue.

Amino acid sequences of hits from mass spectrometry results (PP5-K430R/WT fold change >2) were extracted from UniProt sequence collection (UniProt, 2021) on 05-27-2022. Amino acid sequences were manually searched for p[S/T]P motifs. FDI Lab: SciCrunch Infrastructure antibody registry was searched for antibodies specific to the identified S/TP motifs.

 

## Quantification and statistical analysis

Densitometry was performed using Adobe Photoshop to quantify immunoblot band signal intensity. All statistics were performed using GraphPad Prism version 9.2.0 for macOS (GraphPad Software, La Jolla, California, USA, www.graphpad.com). An unpaired Student's *t*-test was used to determine statistical significance. Significance was denoted as asterisks in Fig. 4D: **$P < 0.01$; ****$P < 0.0001$. Error bars represent the standard deviation (SD) for three independent experiments. No blinding was performed.

## Graphics

Figures 3G and 4F graphics created with BioRender.com.

# Data availability

The datasets produced in this study are available in the following databases: Protein interaction AP-MS data: PRIDE PXD038701 (https://www.ebi.ac.uk/pride/archive/projects/PXD038701).

The source data of this paper are collected in the following database record: biostudies:S-SCDT-10_1038-S44319-024-00250-2.

# Peer review information

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

## Acknowledgements

This work was supported by the National Institute of General Medical Sciences of the National Institutes of Health under Award Number R35GM139584 (MM). The content is solely the responsibility of the authors and does not necessarily represent the official views of the National Institutes of Health. This work was also supported with funds from the SUNY Upstate Medical University and Upstate Foundation. Mass spectrometry analysis performed by Cell Signaling Technologies.

## Author contributions

**Rebecca A Sager**: Data curation; Formal analysis; Methodology; Writing—original draft; Writing—review and editing. **Sarah J Backe**: Resources; Data curation; Formal analysis; Validation; Visualization; Methodology; Writing—original draft; Writing—review and editing. **Diana M Dunn**: Data curation; Methodology. **Jennifer A Heritz**: Data curation; Formal analysis; Methodology. **Elham Ahanin**: Visualization. **Natela Dushukyan**: Data curation; Methodology. **Barry Panaretou**: Conceptualization. **Gennady Bratslavsky**: Conceptualization. **Mark R Woodford**: Conceptualization; Writing—original draft. **Dimitra Bourboulia**: Conceptualization; Writing—original draft. **Mehdi Mollapour**: Conceptualization; Resources; Data curation; Formal analysis; Supervision; Funding acquisition; Validation; Investigation; Methodology; Writing—original draft; Project administration; Writing—review and editing.

Source data underlying figure panels in this paper may have individual authorship assigned. Where available, figure panel/source data authorship is listed in the following database record: biostudies:S-SCDT-10_1038-S44319-024-00250-2.

## Disclosure and competing interests statement

The authors declare no competing interests.

# Expanded View Figures

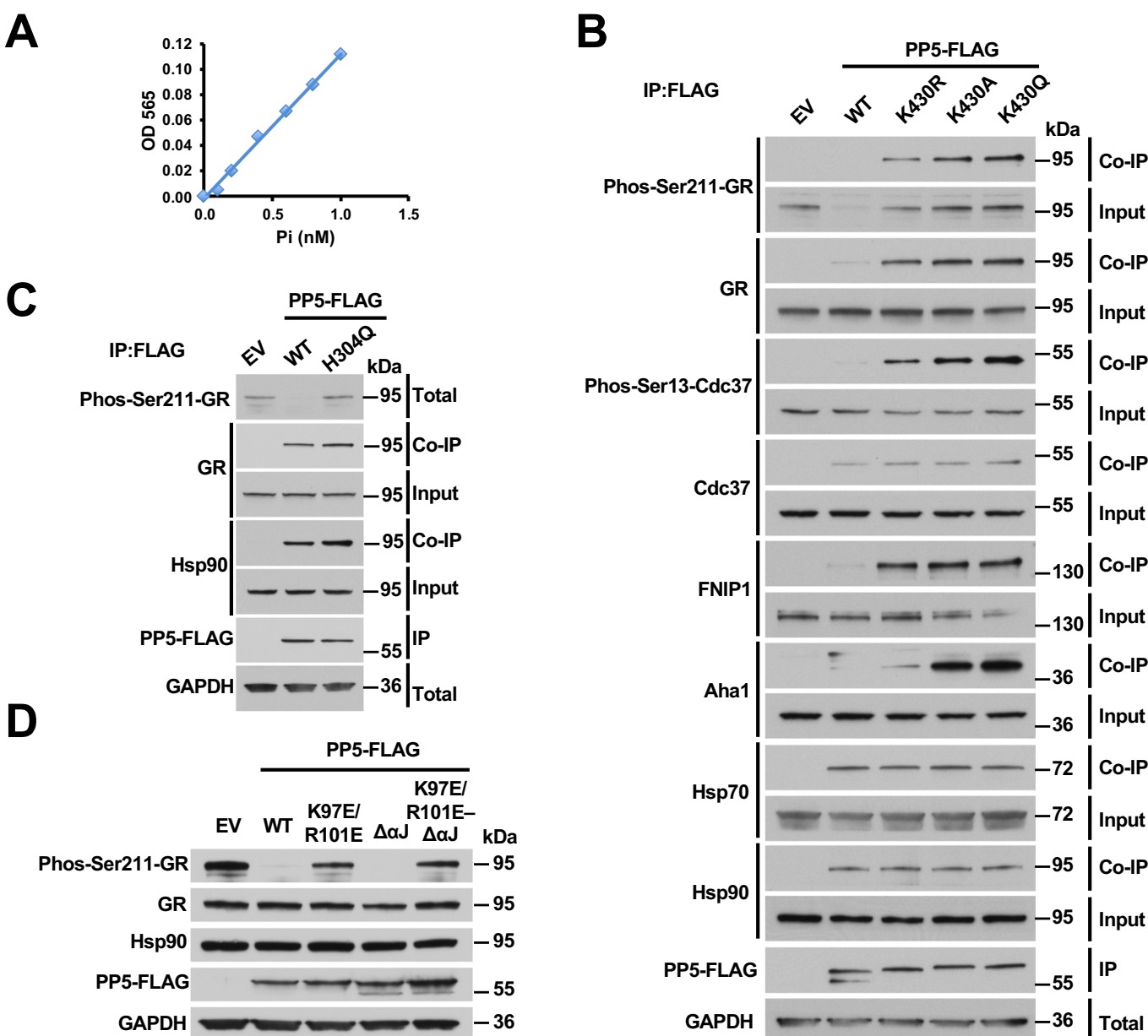

**Figure EV1. Role of PP5 SUMOylation toward phosphatase activity.**

(A) Phosphate standard curve for in vitro PP5 activity measured using PiPer phosphate assay in Fig. 2A. (B) PP5-FLAG WT, K430R, K430A, and K430Q were transiently transfected and IP. Co-IP of chaperones, co-chaperones, and known PP5 substrates were analyzed by immunoblotting. EV was used as a control. (C) Wild-type PP5-FLAG and H304Q mutant were expressed in an IP. Activity and binding were analyzed by Western blot. EV was used as a control. (D) PP5-FLAG WT, K97E/R101E, ΔαJ, and the double mutant PP5-K97E/R101E-ΔαJ were transiently expressed. The activity of PP5, as assessed by dephosphorylation of phospho-GR-S211 was assessed by immunoblotting.

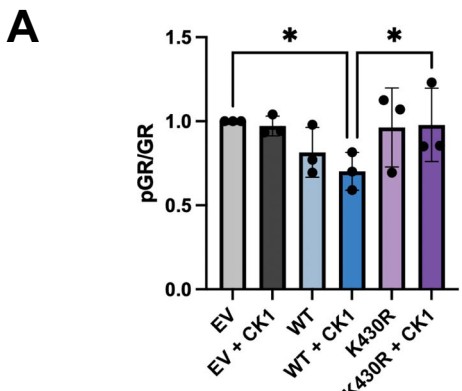

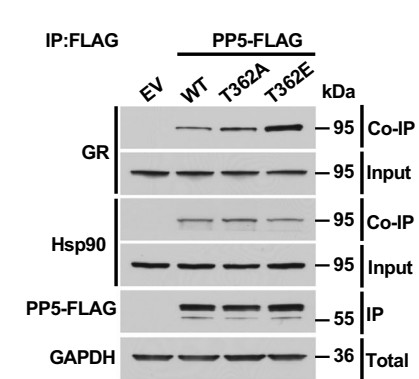

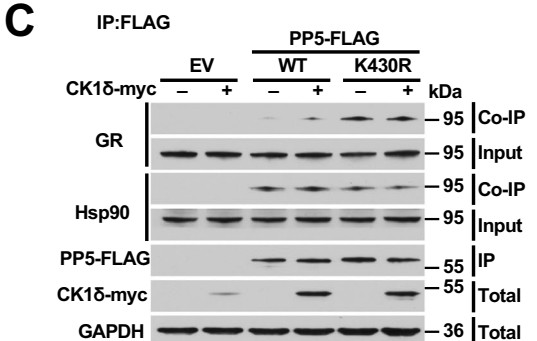

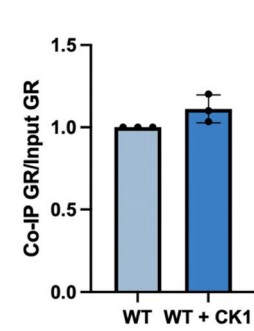

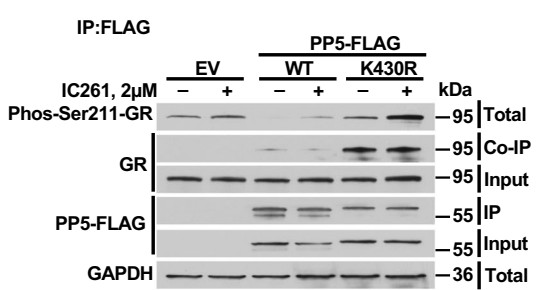

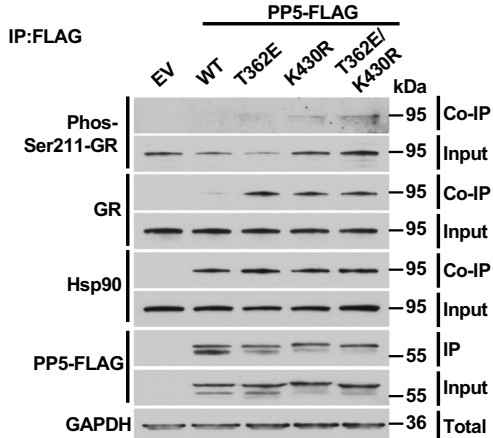

◀ **Figure EV2.  Cross-talk of PP5 phosphorylation and SUMOylation.**

(A) Densitometric analysis of GR-S211 phosphorylation normalized to total GR protein levels from Fig. 3F. Ratios were normalized to EV. Ordinary one-way ANOVA was used to determine statistical significance between samples from three biological replicates ($n = 3$). Data presented show mean ± standard deviation. EV vs. PP5-WT + CK1δ $p = 0.0344$, PP5-WT + CK1δ vs. PP5-K430R + CK1δ $p = 0.0471$. (B) Wild-type PP5-FLAG, PP5-T362A, and PP5-T362E were IP, and the binding of the substrate GR and chaperone Hsp90 were assessed by immunoblotting. (C) PP5-FLAG WT or K430R were co-transfected with EV or CK1δ-myc. Following IP of PP5-FLAG, co-IP of GR, and Hsp90 were assessed by Western blot. EV was used as a control. Samples and representative input blots for GR, Hsp90, CK1δ-myc, and GAPDH are the same as seen in Fig. 3F. (D) Densitometric analysis of GR Co-IP normalized to total GR protein levels from Fig. EV2C. Ratios were normalized to WT. Data were presented as mean ± standard deviation derived from three biologiclal replicates ($n = 3$). (E) HEK293 cells were transfected with EV, PP5-FLAG WT, or PP5-K430R. Cells were then treated with vehicle or 2 μM IC261 (CK1δ inhibitor) for 16 h. PP5 activity and binding to the substrate GR were assessed by immunoblotting. (F) PP5-FLAG WT, T362E, PP5-K430R, and the PP5-T362E-K430R double mutant were IP. The binding of PP5-FLAG mutants to the chaperone Hsp90 and substrate GR were assessed by immunoblotting. EV was used as a control. Samples and representative input blots for phos-S211-GR, GR, Hsp90, PP5-FLAG, and GAPDH are the same as seen in Fig. 3E.

      

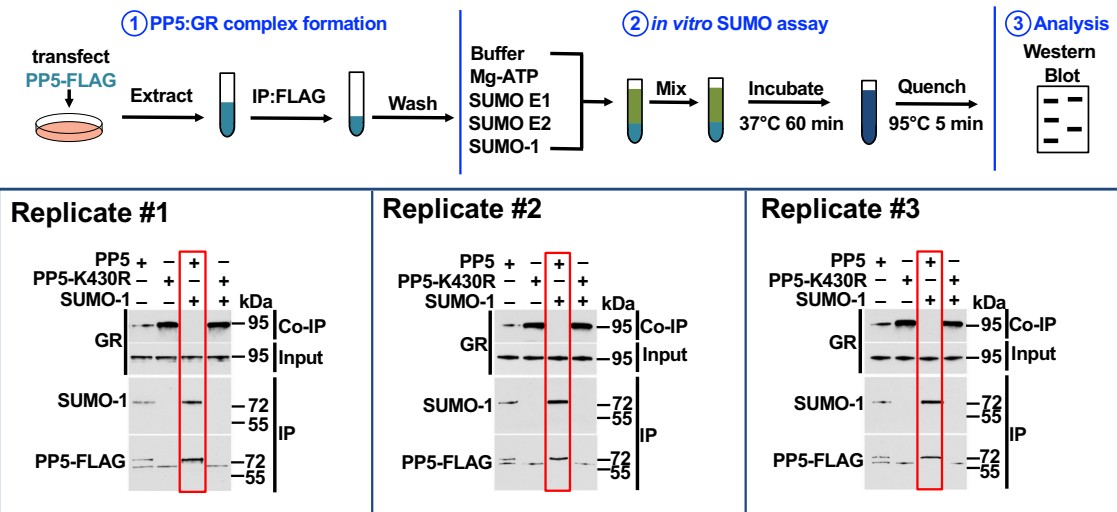

**Figure EV3. PP5 in vitro SUMOylation and substrate release.**

PP5-FLAG WT or K430R were expressed and isolated for use in an in vitro SUMOylation assay. PP5-WT and K430R were incubated with ATP in the presence and absence of recombinant SUMO-1. FLAG IP was used to isolate PP5 following in vitro SUMOylation and interaction with GR was evaluated by immunoblot. PP5 SUMOylation was assessed by immunoblotting. The data shown are three biological replicates. Replicate three is shown in Fig. 4C.

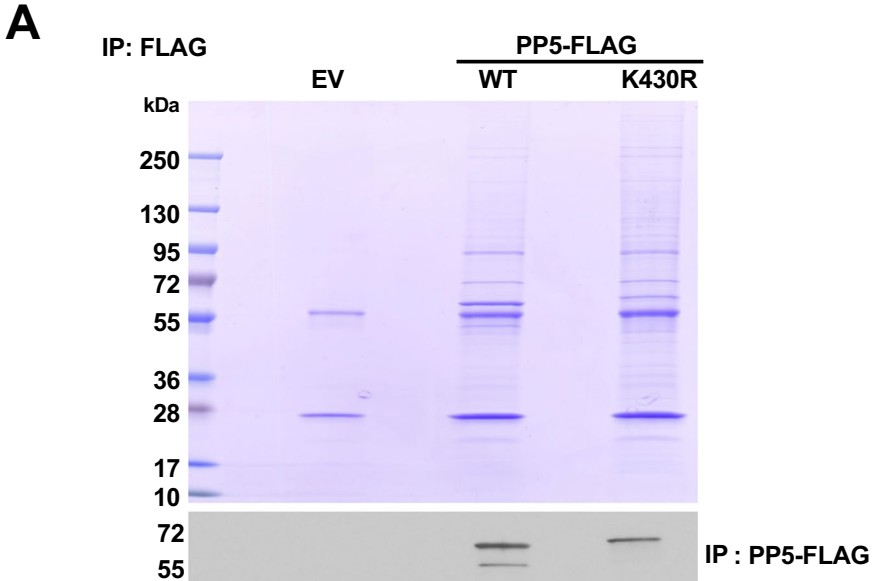

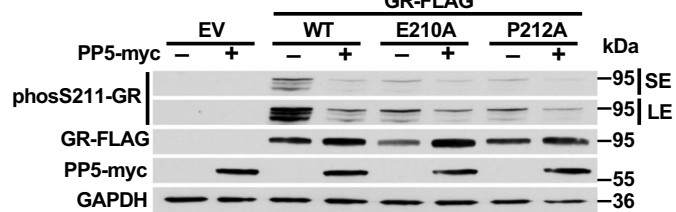

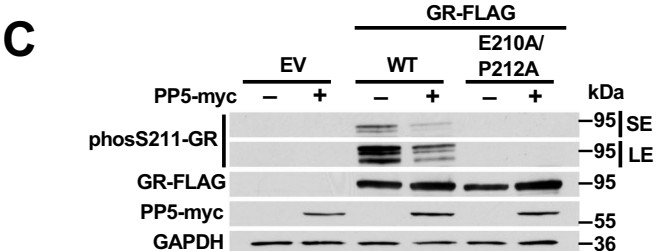

**Figure EV4. Generation and analysis of interactome of PP5-K430R.**

(A) PP5-FLAG WT and K430R were IP from HEK293 cell lysate and subject to mass spectrometry analysis to identify interacting proteins in Fig. 5A. A small aliquot of each sample was also run on SDS-PAGE for examination by Coomassie staining (above) and Western blot for PP5-FLAG (below). EV was used as a control. (B) GR-FLAG WT, GR-E210A, or GR-P212A were transiently transfected with or without co-transfection of PP5-myc. The activity of PP5-FLAG towards dephosphorylation of total phospho-GR-S211 was assessed by immunoblotting. EV was used as a control. (C) GR-FLAG WT or GR-E210A/P212A double mutants were transiently transfected with or without co-transfection of PP5-myc. Activity of PP5-FLAG towards dephosphorylation of phospho-GR-S211 was assessed by immunoblotting. EV was used as a control.

    