## [Peer Review File · EMBO Reports]

SUMOylation of Protein Phosphatase 5 Regulates Phosphatase Activity and Substrate Release

Mehdi Mollapour, Rebecca Sager, Sarah Backe, Diana Dunn, Jennifer Heritz, Elham Ahanin, Natela Dushukyan, Barry Panaretou, Gennady Bratslavsky, Mark Woodford, and Dimitra Bourboulia

Corresponding author(s): Mehdi Mollapour (mollapom@upstate.edu)

Review Timeline:

Submission Date:	10th Apr 24
Editorial Decision:	5th Jul 24
Revision Received:	29th Jul 24
Editorial Decision:	15th Aug 24
Revision Received:	16th Aug 24
Accepted:	20th Aug 24

Transaction Report: The first review round of this manuscript was performed in another journal. Since the original reviews are not subject to EMBO's transparent review process policy, they cannot be published.

Dear Mehdi,

Thank you again for submitting your manuscript to EMBO reports. It has been reviewed at another journal which decided not to offer publication. I had offered to take the anonymous referee comments and your response to these into account in arriving at an editorial decision at EMBO Reports. I note that the major concern raised on your revised manuscript pertained to the in vitro SUMOylation assay and the mechanistic conclusions derived from this experiment (former referee #1 and advisor #4). A second concern raised by former referee #3 pertains to the proposed phosphatase consensus motif and its predictive power. You identified this motif in more than 8,000 proteins and then reduced the number of potential PP5 candidates by comparing to the PhosphoSitePlus database. You reason that PP5 could only act on proteins that are phosphorylated in the first instance. While I agree that this argumentation certainly holds, the predictive power of the consensus motif is limited.

I have asked two experts in the field to act as arbitrator and to comment on your manuscript as well as the previous referees' comments. I received feedback from a chaperone expert who first of all agreed with the limited predictive power of the PP5 consensus motif. The advisor also pointed out that the IP experiment using cell extracts is limited because of the complexity of the Hsp90 system and that in vitro assays would provide cleaner and more convincing evidence for the mechanistic conclusions drawn.

The second advisor with expertise in chaperone biology and the Hsp90 system considered your response adequate within the confines of working with GR in vitro.

Taken together, we have decided to proceed with the publication of your study at EMBO Reports under the condition that you discuss the limitations of the IP/SUMOylation assay in the manuscript and that you tone down conclusions on the PP5 consensus motif and its predictive power.

We now require you to format your manuscript according to the guidelines of EMBO Reports. You will find the general instructions further down in my letter. In addition, I list a few specific points that I noted when going through your manuscript files. Please note that these are only a few points I noted, and that there are other important points in the General instructions, such as Source Data and the Author Checklist.

A) SPECIFIC POINTS:

- Your manuscript will be published as a Report. For short reports, the revised manuscript should not exceed 27,000 characters (including spaces but excluding materials & methods and references) and 5 main plus 5 expanded view figures. The results and discussion sections must further be combined, which will help to shorten the manuscript text by eliminating some redundancy that is inevitable when discussing the same experiments twice.
- Regarding the Author Contributions, we now use CRediT to specify the contributions of each author in the journal submission system. Therefore, please remove the Author Contributions from the manuscript file and make sure that the author contributions in our online manuscript tracking system are correct and up-to-date. The information you specified in the system will be automatically retrieved and typeset into the article. You can enter additional information in the free text box provided, if you wish.
- "DECLARATION OF INTERESTS" should be called 'Disclosure and competing interests statement'. For more information see <https://www.embopress.org/page/journal/14693178/authorguide#conflictsofinterest>
- Please remove the INCLUSION AND DIVERSITY STATEMENT. You might want to include it in the Acknowledgment section, but we unfortunately do not have such a paragraph in our manuscripts.
- Fig 2A shows the mean of duplicate experiments. If $n < 2$, please show only the individual datapoints and not the mean. If $n > 3$ you can show the mean but the graphs must also include the individual datapoints.
- Please reformat the Reagents and Resource table to our format and upload it as an individual file choosing the file type "Reagents and Tools table". You can download our template here: <https://www.embopress.org/page/journal/14693178/authorguide#structuredmethods>.
- You note in the methods that plasmids are available on request but please also note our editorial policies that all published materials need to be available to the community unless good reasons apply.
- Data and Code Availability is called "Data Availability", listed at the end of the Methods section, and it needs an URL that resolves directly to the dataset on PRIDE. Please only keep the statement and link to the deposited dataset (plus URL) and remove all other text. It should look like this:

Data availability

- [data type]: [name of the resource] [accession number/identifier/doi] ([URL or identifiers.org/DATABASE:ACCESSION])
- References should have no DOI in the reference list.
- Supplementary files are either an Appendix PDF or EV figures/tables (see general instructions below).
- Table S1 should be converted to Dataset EV1 and uploaded as file type 'Dataset'.

B) GENERAL FORMATTING GUIDELINES:

2) individual production quality figure files as .eps, .tif, .jpg (one file per figure). Please download our Figure Preparation Guidelines (figure preparation pdf) from our Author Guidelines pages <https://www.embopress.org/page/journal/14693178/authorguide> for more info on how to prepare your figures.

4) a complete author checklist, which you can download from our author guidelines (<<https://www.embopress.org/page/journal/14693178/authorguide>>). Please insert information in the checklist that is also reflected in the manuscript. The completed author checklist will also be part of the RPF.

5) Please note that all corresponding authors are required to supply an ORCID ID for their name upon submission of a revised manuscript (<<https://orcid.org/>>). Please find instructions on how to link your ORCID ID to your account in our manuscript tracking system in our Author guidelines (<<https://www.embopress.org/page/journal/14693178/authorguide#authorshipguidelines>>)

6) We replaced Supplementary Information with Expanded View (EV) Figures and Tables that are collapsible/expandable online. A maximum of 5 EV Figures can be typeset. EV Figures should be cited as 'Figure EV1, Figure EV2' etc... in the text and their respective legends should be included in the main text after the legends of regular figures.

7) Please note that a Data Availability section at the end of Materials and Methods is now mandatory. In case you have no data that requires deposition in a public database, please state so instead of refereeing to the database. See also < <https://www.embopress.org/page/journal/14693178/authorguide#dataavailability>>. Please note that the Data Availability Section is restricted to new primary data that are part of this study.

Additional information on source data and instruction on how to label the files are available <<https://www.embopress.org/page/journal/14693178/authorguide#sourcedata>>.

10) Figure legends and data quantification:

- the name of the statistical test used to generate error bars and P values,
 - the number (n) of independent experiments (please specify technical or biological replicates) underlying each data point,
 - the nature of the bars and error bars (s.d., s.e.m.)
-
- If the data are obtained from n {less than or equal to} 5, show the individual data points in addition to the SD or SEM.
 - If the data are obtained from n {less than or equal to} 2, use scatter blots showing the individual data points.

11) Our journal encourages inclusion of *data citations in the reference list* to directly cite datasets that were re-used and obtained from public databases. Data citations in the article text are distinct from normal bibliographical citations and should directly link to the database records from which the data can be accessed. In the main text, data citations are formatted as follows: "Data ref: Smith et al, 2001" or "Data ref: NCBI Sequence Read Archive PRJNA342805, 2017". In the Reference list, data citations must be labeled with "[DATASET]". A data reference must provide the database name, accession number/identifiers and a resolvable link to the landing page from which the data can be accessed at the end of the reference. Further instructions are available at <<https://www.embopress.org/page/journal/14693178/authorguide#referencesformat>>.

12) All Materials and Methods need to be described in the main text using our 'Structured Methods' format, which is required for all research articles. According to this format, the Methods section includes a Reagents and Tools Table (listing key reagents, experimental models, software and relevant equipment and including their sources and relevant identifiers) followed by a Methods and Protocols section describing the methods using a step-by-step protocol format. The aim is to facilitate adoption of the methodologies across labs. More information on how to adhere to this format as well as a downloadable template (.docx) for the Reagents and Tools Table can be found in our author guidelines:

13) As part of the EMBO publication's Transparent Editorial Process, EMBO Reports publishes online a Review Process File to accompany accepted manuscripts. This File will be published in conjunction with your paper and will include the referee reports, your point-by-point response and all pertinent correspondence relating to the manuscript.

Kind regards,

Martina

Referee #2:

The authors have adequately responded to prior reviews and the current version of their manuscript nicely demonstrates (within the confines of the challenging experimental limitations of working with GR in vitro) that SUMOylation promotes substrate release. As the manuscript stands, it represents an important contribution to the field.

Point-by-point response to Reviewers' comments: April 4, 2024

Reviewer #1: (Comments for the author)

I am not convinced by the *in vitro* assay which is a pivotal experiment to support the claims. How do the author explain that adding sumo has any effect whilst sumoylation requires a series of enzymatic steps.

We would like to clarify that we did not just add "SUMO". As we stated clearly in our previous responses, the figure legend, the results section and the methods section that we used an *in vitro* SUMOylation assay (Enzo Cat# BML-UW8955-0001). This widely accepted assay kit contains all the necessary enzymes, SUMO-1 and reagents for enzymatic SUMOylation to occur.

Again, we would like to reiterate that PP5-FLAG or non-SUMO mutant PP5-FLAG-K430R were initially immunoprecipitated (lanes 1 & 2- Figure 4C)) and then they were subject to *in vitro* SUMOylation (lanes 3 & 4- Figure 4C). This process ensures that we isolated the physiological PP5:GR complex from cells and allowed us to determine the effect of SUMOylation on the preformed complex.

We have now made a more comprehensive schematic for **Figure 4C** to detail each step of the assay kit. Furthermore, we have clarified this point in the figure legend. We are more than happy to replace Figure 4C in our manuscript with the version below.

Reviewer #3: The authors now substantially addressed reviewers' concerns and removed ambiguous statements.

It remains a merit of the study that the authors describe the pathway phosphorylation at T362 triggers SUMOylation at K430, and this is linked to substrate release. These are substantial mechanistic insights.

The authors make in some of the answers the point that there is a confusion between binding and dephosphorylation motifs. Key is that dephosphorylation requires an interaction at an motif, which you may as well call binding. Anyway, the important bit of this is the selectivity of the dephosphorylation. A generic motif present in all proteins is not specific for regulation. This was the background of the question to the authors. Thus, one minor point remains:

Please add an interpretation of the specificity to the discussion. We have 20,000 genes.

a. What does it mean of 7780 or 2130 can be phosphorylated?

We appreciate the reviewer giving us a chance to clarify this point. The reviewer previously asked 'how many proteins in the human genome would have the motif in question'. The answer to this question was shown in Table X (8,094 genes encode an S-P motif). However, to be considered a PP5 substrate, PP5 would need to dephosphorylate the protein within this motif. This means the protein must first be phosphorylated on this motif. Therefore, we searched the PhosphoSitePlus Site Search tool to for phosphorylation events fitting our proposed consensus motif. This resulted in 7,780 proteins that are phosphorylated on a serine preceding a proline. This suggests that ~38% of proteins encoded in the genome have potential to be PP5 substrates.

b. What re the implications of this for the regulation of the pathway?

It is well established that there are more protein kinases than protein phosphatases. As such, phosphatases are known to regulate disparate cellular pathways. Specifically in regards to PP5, we and others have shown that PP5 is involved in cell survival (Dushukyan *et al.*, 2017, *Cell Rep*), apoptosis (Ahanin *et al.*, 2023, *Cell Chem Biol*), DNA damage response (Wechsler *et al.* 2004, *Proc Natl Acad Sci*), and regulation of steroid hormone receptors (Kaziales, A. *et al.*, 2020, *Sci Rep*; Sager *et al.*, 2020, *Cell Stress Chaperones*). We agree with the reviewer that PP5-SUMOylation likely provides an additional layer of regulation to one or more of these pathways, however, deciphering how SUMOylation of PP5 affects each downstream pathway is outside the scope of this paper.

Reviewer #4:

- The *in vitro* SUMOylation experiment needs to be done with purified proteins produced in bacteria as the pull down system is not well controlled.

We appreciate this comment from the reviewer and in fact, this was our initial plan for examining *in vitro* SUMOylation of PP5 leading to release of substrate glucocorticoid receptor (GR). However, there are many technical challenges at each stage of the experiment rendering this idea impossible to execute. Firstly, PP5:GR complex assembly requires many factors including (e.g. chaperone proteins). Seminal works by the Pratt, Smith and Toft groups have extensively shown the complexity of reconstituting

GR complexes *in vitro* (Bresnick *et al.*, 1989, *J Biol Chem*; Dittmar *et al.*, 1997, *J Biol Chem*; Johnson and Toft, 1994, *J Biol Chem*; Picard *et al.*, 1990, *Nature*; Pratt and Toft, 1997, *Endocr Rev*; Pratt *et al.*, 1992, *J Steroid Biochem Mol Biol*; Pratt and Toft, 2003, *Exp Biol Med*; Johnson *et al.*, 2000, *J Biol Chem*). Secondly and perhaps most importantly, it is impossible to express full-length GR in bacteria and retain functionality. This has been shown by the pioneering experts in the field of GR and chaperones; Pratt, Smith and Toft, mentioned above.

We also considered using purified full-length recombinant GR from insect cells available from Invitrogen. However, they state the following in their product information sheet;

*'Glucocorticoid receptor recombinant protein is an insect cell-expressed, full-length, untagged, human nuclear hormone receptor that is **partially purified** in order to maintain high ligand binding activity. **Further purification results in decreased activity, presumably due to loss of endogenous accessory proteins.**'*

Given these issues, we instead isolated WT-PP5:GR complexes (containing SUMOylated and non-SUMOylated PP5) and K430R-PP5:GR complexes (representing only non-SUMO PP5). We then carried out the enzymatic SUMOylation steps of the commercially-available kit *in vitro* SUMO kit (Enzo Life Sciences) and presented the data in Figure 4C. Our data is robust and clearly demonstrates that SUMOylation of PP5 cause dissociation of GR substrate.

Based on the issues listed above and our experience studying protein SUMOylation (Mollapour *et al.*, 2014 *Molecular Cell*), we believe that this is the only way to conduct the *in vitro* experiment and obtain a meaningful and physiologically relevant data.

- SUMOylation of PP5 must be shown more convincingly under endogenous conditions. Respectfully, we have provided evidence of PP5-SUMOylation under endogenous conditions in Figure 1C. We have immunoprecipitated *endogenous* PP5 and showed, the SUMOylation of PP5 in two ways (immunoblotting with anti-SUMO antibody, and molecular weight shift). We do not quite understand what “more” convincingly would entail as no specific experiment was proposed by the reviewer.

References

- Ahanin EF, Sager RA, Backe SJ, Dunn DM, Dushukyan N, Blanden AR, Mate NA, Suzuki T, Anderson T, Roy M, Oberoi J, Prodromou C, Nsouli I, Daneshvar M, Bratslavsky G, Woodford MR, Bourboulia D, Chisholm JD, Mollapour M. Catalytic inhibitor of Protein Phosphatase 5 activates the extrinsic apoptotic pathway by disrupting complex II in kidney cancer. *Cell Chem Biol.* 2023 Oct 19;30(10):1223-1234.e12. doi: 10.1016/j.chembiol.2023.06.026. Epub 2023 Jul 31. PMID: 37527661; PMCID: PMC10592443.
- Bresnick EH, Dalman FC, Sanchez ER, Pratt WB. Evidence that the 90-kDa heat shock protein is necessary for the steroid binding conformation of the L cell glucocorticoid receptor. *J Biol Chem.* 1989 Mar 25;264(9):4992-7. PMID: 2647745.
- Dittmar KD, Pratt WB. Folding of the glucocorticoid receptor by the reconstituted Hsp90-based chaperone machinery. The initial hsp90.p60.hsp70-dependent step is sufficient for creating the steroid binding conformation. *J Biol Chem.* 1997 May 16;272(20):13047-54. doi: 10.1074/jbc.272.20.13047. PMID: 9148915.
- Dushukyan N, Dunn DM, Sager RA, Woodford MR, Loiselle DR, Daneshvar M, Baker-Williams AJ, Chisholm JD, Truman AW, Vaughan CK, Haystead TA, Bratslavsky G, Bourboulia D, Mollapour M. Phosphorylation and Ubiquitination Regulate Protein Phosphatase 5 Activity and Its Prosurvival Role in Kidney Cancer. *Cell Rep.* 2017 Nov 14;21(7):1883-1895. doi: 10.1016/j.celrep.2017.10.074. PMID: 29141220; PMCID: PMC5699234.
- Johnson JL, Toft DO. A novel chaperone complex for steroid receptors involving heat shock proteins, immunophilins, and p23. *J Biol Chem.* 1994 Oct 7;269(40):24989-93. PMID: 7929183.
- Kaziales A, Barkovits K, Marcus K, Richter K. Glucocorticoid receptor complexes form cooperatively with the Hsp90 co-chaperones Pp5 and FKBP. *Sci Rep.* 2020 Jul 1;10(1):10733. doi: 10.1038/s41598-020-67645-8. PMID: 32612187; PMCID: PMC7329908.
- Picard D, Khursheed B, Garabedian MJ, Fortin MG, Lindquist S, Yamamoto KR. Reduced levels of hsp90 compromise steroid receptor action in vivo. *Nature.* 1990 Nov 8;348(6297):166-8. doi: 10.1038/348166a0. PMID: 2234079.
- Pratt WB, Toft DO. Steroid receptor interactions with heat shock protein and immunophilin chaperones. *Endocr Rev.* 1997 Jun;18(3):306-60. doi: 10.1210/edrv.18.3.0303. PMID: 9183567.
- Pratt WB, Scherrer LC, Hutchison KA, Dalman FC. A model of glucocorticoid receptor unfolding and stabilization by a heat shock protein complex. *J Steroid Biochem Mol Biol.* 1992 Mar;41(3-8):223-9. doi: 10.1016/0960-0760(92)90348-m. PMID: 1373296.
- Pratt WB, Toft DO. Regulation of signaling protein function and trafficking by the hsp90/hsp70-based chaperone machinery. *Exp Biol Med (Maywood).* 2003 Feb;228(2):111-33. doi: 10.1177/153537020322800201. PMID: 12563018.
- Sager RA, Dushukyan N, Woodford M, Mollapour M. Structure and function of the co-chaperone protein phosphatase 5 in cancer. *Cell Stress Chaperones.* 2020 May;25(3):383-394. doi: 10.1007/s12192-020-01091-3. Epub 2020 Apr 2. PMID: 32239474; PMCID: PMC7193036.
- Wechsler T, Chen BP, Harper R, Morotomi-Yano K, Huang BC, Meek K, Cleaver JE, Chen DJ, Wabl M. DNA-PKcs function regulated specifically by protein phosphatase 5. *Proc Natl Acad Sci U S A.* 2004 Feb 3;101(5):1247-52. doi: 10.1073/pnas.0307765100. Epub 2004 Jan 20. PMID: 14734805; PMCID: PMC337038.

Dr Martina Rembold
Senior Scientific Editor
EMBO Reports

Mehdi Mollapour PhD
750 East Adams St
Syracuse, NY 13210
Phone: +1-315-464-8749
mollapom@upstate.edu
www.mollapourlab.com

July 29, 2024

EMBOR-2024-59387V2

Dear Dr Rembold,

Thank you for provisionally accepting our manuscript by Sager et al., entitled "SUMOylation of Protein Phosphatase 5 Regulates Phosphatase Activity and Substrate Release" for publication in *EMBO Rep.*

We have addressed both the specific points and general formatting requirements for our manuscript. We have also followed your advice and discussed the limitations of the immunoprecipitation/SUMOylation assays in the manuscript. We have additionally toned down our conclusions related to the PP5 consensus motif and its predictive power. We used track changes to make these corrections in our manuscript. Please see below the excerpts of these changes;

"The notable limitation of this assay is that there may be additional cellular components present in the PP5:GR complex which could potentially contribute to the substrate release."

"It is likely that additional substrate specificity is driven by an interaction motif separate from the dephosphorylation motif, such as the generalized TPR binding motif (EEVD) that was recently identified (Devi et al, 2024). This EEVD-like binding motif in combination with our proposed dephosphorylation motif may help in identification of new candidate substrate residues in other known interactors."

Per your request, we have significantly shortened our manuscript to a Report containing 24,501 character-count with spaces. It also includes 5 Figures, and 4 expanded views. All authors have agreed to the manuscript content, including the data interpretation and presentation. The authors do not have any financial interests or conflicts to report. We certify that this submission represents original work and is not under review at any other publication.

Once again, thank you for considering our manuscript for publication in *EMBO Rep.*

Sincerely,

Mehdi Mollapour PhD
Professor of Urology, Biochemistry and Molecular Biology
Vice Chair for Translational Research, Department of Urology
SUNY Upstate Medical University

Manuscript number: EMBOR-2024-59387V2

Title: SUMOylation of Protein Phosphatase 5 Regulates Phosphatase Activity and Substrate Release

Author(s): Mehdi Mollapour, Rebecca Sager, Sarah Backe, Diana Dunn, Jennifer Heritz, Elham Ahanin, Natela Dushukyan, Barry Panaretou, Gennady Bratslavsky, Mark Woodford, and Dimitra Bourboulia

Dear Mehdi,

Thank you for the submission of your revised manuscript to EMBO Reports. I had a final look at all the files and am now writing with an 'accept in principle' decision, which means that I will be happy to accept your manuscript for publication once a few minor issues/corrections have been addressed, as follows.

- Please reduce the number of keywords to 5.
 - Funding information in the online manuscript tracking system: SUNY Upstate Medical University and Upstate Foundation should be removed from the Comments box and need to be inserted as separate funder(s).
 - MATERIALS and METHODS should be METHODS
 - BioRender should be acknowledged at the end of the Methods section in the following way:
Graphics:
(some of the... OR Figure #... OR synopsis) Graphics were created with BioRender.com.
 - You show the blot from Figure 4C again in Figure EV3 (Replicate 3). While this is fine and we decided to showcase reproducibility, please clearly state the reuse in the respective figure legends.
 - Our production/data editors have asked you to clarify several points in the figure legends (see below). Please incorporate these changes in the manuscript and return the revised file with tracked changes with your final manuscript submission.
- A) Statistical test information. Only p-values that are actually shown in the figure panel(s) should (and must) be defined in the legends, all others should be removed from (or added to) the legend. Moreover, we ask for the specification of exact p-values:
- Please note that the exact p values are not provided in the legends of figures 4d; EV 2a.

B) Replicates and error bars:

- Please note that the error bars are not defined in the legend of figure EV 2a.
- As a standard procedure we edit the title and abstract. Please find my suggestion (with minor modifications) below my signature.
- Finally, EMBO Reports papers are accompanied online by
 - A) a short (1-2 sentences) summary of the findings and their significance,
 - B) 2-3 bullet points highlighting key results and
 - C) a schematic summary figure that provides a sketch of the major findings (not a data image).Please provide the summary figure as a separate file in PNG or JPG format at a size of 550x300-600 pixels (width x height). Please note that the size is rather small and that text needs to be readable at the final size. Please send us this information along with the revised manuscript.

Once you have made these minor revisions, please use the following link to submit your corrected manuscript:

Link Not Available

If all remaining corrections have been attended to, you will then receive an official decision letter from the journal accepting your manuscript for publication in the next available issue of EMBO reports. This letter will also include details of the further steps you need to take for the prompt inclusion of your manuscript in our next available issue.

Thank you for your contribution to EMBO reports.

Kind regards,

Martina

Martina Rembold, PhD

Abstract

The serine/threonine protein phosphatase 5 (PP5) regulates hormone and stress-induced signaling networks. Unlike other phosphoprotein phosphatases, PP5 contains both regulatory and catalytic domains and is further regulated through post-translational modifications (PTMs). Here we identify that SUMOylation of K430 in the catalytic domain of PP5 regulates phosphatase activity. Additionally, phosphorylation of PP5-T362 is pre-requisite for SUMOylation, suggesting the ordered addition of PTMs regulates PP5 function in cells. Using the glucocorticoid receptor, a well known substrate for PP5, we demonstrate that SUMOylation results in substrate from PP5. We harness this information to create a non-SUMOylatable K430R mutant as a 'substrate trap' and globally identified novel PP5 substrate candidates. Lastly, we generated a consensus dephosphorylation motif using known substrates, and verified its presence in the new candidate substrates. This study unravels the impact of cross talk of SUMOylation and phosphorylation on PP5 phosphatase activity and substrate release in cells.-

All editorial and formatting issues were resolved by the authors.

Dr. Mehdi Mollapour
SUNY Upstate Medical University
Urology
750 East Adams Street
Syracuse, NY 13210
United States

Dear Dr. Mollapour,

I am very pleased to accept your manuscript for publication in the next available issue of EMBO reports. Thank you for your contribution to our journal.

Kind regards,
